# X-ray nanotomography of coccolithophores reveals that coccolith mass and segment number correlate with grid size

T. Beuvier [1,2], I. Probert[3], L. Beaufort[4], B. Suchéras-Marx[4], Y. Chushkin[2], F. Zontone[2] & A. Gibaud[1]

Coccolithophores of the Noëlaerhabdaceae family are covered by imbricated coccoliths, each composed of multiple calcite crystals radially distributed around the periphery of a grid. The factors that determine coccolith size remain obscure. Here, we used synchrotron-based three-dimensional Coherent X-ray Diffraction Imaging to study coccoliths of 7 species of *Gephyrocapsa*, *Emiliania* and *Reticulofenestra* with a resolution close to 30 nm. Segmentation of 45 coccoliths revealed remarkable size, mass and segment number variations, even within single coccospheres. In particular, we observed that coccolith mass correlates with grid perimeter which scales linearly with crystal number. Our results indirectly support the idea that coccolith mass is determined in the coccolith vesicle by the size of the organic base plate scale (OBPS) around which R-unit nucleation occurs every 110–120 nm. The curvature of coccoliths allows inference of a positive correlation between cell nucleus, OBPS and coccolith sizes.

[1] LUNAM, IMMM, UMR 6283 CNRS, Faculté des Sciences, 72085 Le MANS Cedex 09, France. [2] European Synchrotron Radiation Facility, 71, avenue des Martyrs, 38000 Grenoble, France. [3] Sorbonne Université / CNRS, Roscoff Culture Collection, FR2424, Station Biologique de Roscoff, Place Georges Teissier, 29680 Roscoff, France. [4] Aix Marseille Univ, CNRS, IRD, INRA, Coll France, CEREGE, Aix-en-Provence, France. Correspondence and requests for materials should be addressed to T.B. (email: tbeuvier@yahoo.fr) or to A.G. (email: alain.gibaud@univ-lemans.fr)

Coccolithophores are unicellular marine planktonic algae that have significantly impacted global biogeochemical cycles since their appearance around 220 million years ago, both via their ability to produce organic carbon (photosynthesis) and inorganic carbon (calcification). Coccolithophores produce calcareous scales (coccoliths) inside the cell that are subsequently extruded to form an exoskeleton, termed a coccosphere. Despite their minute (micrometer-scale) size, coccoliths are biogeochemically relevant because coccolithophores are one of the most abundant phytoplankton groups in the marine environment, to such an extent that these unicellular algae are considered being the main extant calcifying organisms with a coccolith production of ~$10^{26}$ year$^{-1}$ [1–5].

Therefore understanding how environmental factors influence the degree of calcification of coccoliths is of significant interest. In particular the main current focus concerns the potential impact of ocean acidification. At present, the ocean absorbs about one-third of atmospheric $CO_2$ emissions, resulting in a shift in seawater carbonate chemistry: while dissolved $CO_2$ and $HCO_3^-$ concentrations increase, pH, $CO_3^{2-}$ concentration and calcium carbonate saturation states ($\Omega$) decrease[6,7]. This global ocean change affects a variety of marine life forms. In particular, ocean acidification has been reported to decrease the degree of biogenic calcification of corals[8,9], echinoderms[10], and foraminifera[11,12]. However, despite their crucial role in ocean, the response of coccolithophore calcification to ocean acidification is far from being well understood[13–17]. One of the reasons for this is the technical difficulty of measuring the mass of individual coccoliths which weighs <400 pg[18], with coccoliths produced by members of the most widespread extant coccolithophore family, the Noëlaerhabdaceae (including the genera Gephyrocapsa, Emiliania, and Reticulofenestra), being particularly light (i.e., mass ranging from 1 to 30 pg).

Average coccolith and coccosphere mass can be obtained in bulk samples from measurements of total calcite mass and total cell concentration[18–24], but measuring the mass of individual coccoliths is more challenging. Mass can be estimated from in-plane and out-of-plane 2D images[18] or from resonance frequency difference as demonstrated for a quite large (~12 μm) coccolith of Coccolithus pelagicus[25]. However, the only technique to date able to determine the mass of large quantities of individual coccoliths is polarized light microscopy (PLM) which deduces the in-plane thickness of individual coccoliths from the brightness observed in circular or cross polarization[17,26–29].

Three-dimensional X-ray coherent diffraction imaging (3D-CXDI) is a novel synchrotron technique based on Fourier transformation of a numerically-phased coherent scattering pattern oversampled in the far-field which provides the 3D electron density distribution of isolated objects. The spatial resolution of 3D-CXDI is currently < 100 nm[30,31], i.e., intermediate between optical and electron microscopy, hence the technique is well suited to image coccospheres of 1–7 μm size and to determine the mass of individual coccoliths. Moreover, 3D-CXDI has the key advantage compared to PLM of being independent of the c-axis orientation of calcite crystals. In addition, 3D-CXDI can access not only the thickness of coccoliths but all morphological features of these calcareous plates even within a single coccosphere.

The mass of coccoliths varies considerably between and within species[13,27]. Amongst the main environmental variables, carbonate chemistry[13], alkalinity[17], and salinity[32], as well as phosphate level[33–35] and trace metal concentration[36] of seawater may affect the calcification of coccoliths. However, during their formation coccoliths are not in direct contact with seawater but are rather formed inside the cell, generally one at a time, in a distinct compartment called the coccolith vesicle (CV)[37,38]. The proximal side of the CV is closely apposed to the nuclear membrane and the distal side is intimately associated with a reticular body (RB) in E. huxleyi and G. oceanica[39–41] and probably also in R. parvula. Coccolithogenesis within the CV begins with the formation of a protococcolith ring at the periphery of an underlying organic base-plate scale (OBPS)[38,40,42,43]. In other words, the perimeter of the OBPS may be similar to that of the peripheral grid, which is constructed by the intersection between the tube and the grid (called also "central area"). By assuming that the peripheral grid perimeter is an ellipse, the measurement of the major axis $a_g$ and the minor axis $b_g$ of the grid allows determination of $p$ from $p = \pi \sqrt{(a_g{}^2 + b_g{}^2)/2 - (a_g - b_g)^2/8}$. By using X-ray tomography to image coccolithophores in three dimensions, our work highlights that the mass $m$ and the segment number $n$ of noëlaerhabdacean coccoliths correlate with grid perimeter $p$ according to $m \sim p^{3.125}$ and $n \sim p$. In particular, the proportionality between $n$ and $p$ means that the width of the segments is a constant close to 110–120 nm, whatever the coccolithophore species. As the grid size of mature coccoliths is related to the OBPS size around which nucleation and growth of the coccoliths took place, we propose that the outer perimeter of the OBPS fixes the $CaCO_3$ nucleation site number (with one site every 110–120 nm) and as a consequence the segment number. We therefore speculate that the large variability in mass and segment number of coccoliths in a single coccosphere may originate from the variability in size of the OBPS during the cell growth/division cycle.

## Results

**X-ray nanotomography of coccolithophores**. Details about the principle of CXDI, the real resolution and the analysis methods are given in Supplementary Note 1 and Supplementary Figs. 1–5. Comparisons between SEM and 3D-CXDI images validated the accuracy of 3D-CXDI reconstructions as shown for G. oceanica in Fig. 1a, b. Additional 3D-CXDI reconstructions for other noëlaerhabdaceaen species are displayed in Fig. 1c. The coccospheres had external diameters ranging from $\Phi_{CS} = 4.1$ μm for G. ericsonii to $\Phi_{CS} = 7.3$ μm for G. oceanica and contained between $C_N = 10$ and 24 coccoliths per coccosphere. As reported for other coccolithophore species[44,45], the larger the coccosphere, the longer the coccolith major axis (Supplementary Fig. 6). E. huxleyi RCC1212 was the only culture strain that had a coccosphere with two layers of coccoliths (Supplementary Fig. 7).

**From coccospheres to coccoliths**. Individual coccoliths were extracted from 3D matrices of the coccospheres using manual segmentation of the 3D array via the ImageJ software (Supplementary Fig. 3). Top distal views show the elliptical shape of coccoliths with shield excentricities ranging from 0.48 to 0.65, whereas central area eccentricities range from 0.66 to 0.81 (Fig. 2a, Supplementary Fig. 8, Supplementary Table 1). These eccentricities may originate from the elliptical shape of the OBPS on/around which the nucleation of the coccolith took place. In addition, cross-sections show that both shields of the coccoliths are out-of-plane inclined by about $\alpha \sim 30 \pm 5°$ along the major axis and $\alpha \sim 25 \pm 5°$ along the minor axis (Fig. 2b, c). These inclinations likely correspond to the curvature of the nuclear membrane to which the coccolith vesicle is apposed during intracellular formation of the coccolith[40,41]. Whereas the major axis of the coccolith varies from 2 to 6 μm between species, the constant value of the inclination of the shield $\alpha$ suggests a positive correlation between the size of the cell nucleus and coccolith size. When coccospheres are mechanically deformed on contact with the $Si_3N_4$ support, some coccoliths may exhibit an increase or a decrease of the out-of-plane inclination which in this case is an

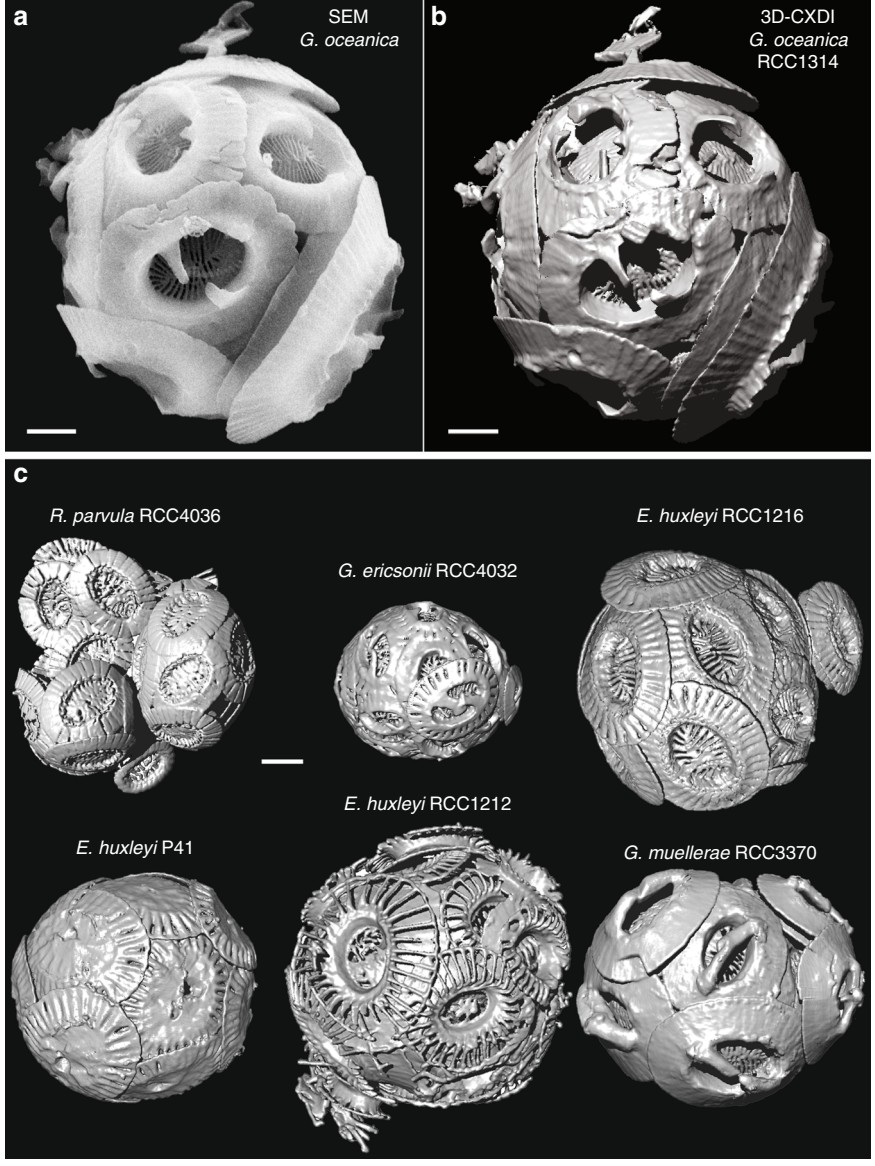

**Fig. 1** 3D-CXDI of coccospheres. **a** SEM image of *G. oceanica RCC1314*. **b** 3D-CXDI view of *G. oceanica RCC1314*. **c** 3D-CXDI views of six other coccospheres. Scale bar = 1 μm

experimental artifact (Supplementary Fig. 9). The top view thicknesses of the coccoliths are shown in Fig. 2d. They are calculated by multiplying the cubic root of one voxel (i.e., 28.8 or 32.5 nm) by the number of voxels along the direction normal to the plane of the coccolith having an intensity greater than the isovalue. The isovalue is the voxel intensity below which a voxel is considered as empty and above which it is considered as filled by calcium carbonate. For comparison, the top view images of individual coccoliths obtained by PLM are shown in Fig. 2e. Both the X-ray and optical images clearly show that thickness is maximal in the tube region of coccoliths. As the tube region is in direct contact with the central area, this observation highlights in a qualitative way the link between the size of the central area and the mass of the coccoliths.

**Mass of coccoliths on single coccospheres**. For a quantitative approach, the volume of individual extracted coccoliths was determined from 3D-CXDI and converted to mass by assuming that the density of calcite is 2.71 g/cm$^3$. The uncertainties in volume measurements are detailed in the supplementary

information (Supplementary Note 1 and Supplementary Figs. 4 and 5) and the results are summarized in Fig. 3 (black dots). The mass of coccoliths significantly varied in single coccospheres. For instance, the coccosphere of *G. oceanica* analyzed by 3D-CXDI was composed of 10 coccoliths, the mass of which varied more than threefold ($m = 7.2–23.1$ pg). A similar variability was observed for *E. huxleyi RCC1212* ($m = 1.5–5.1$ pg). 3D-CXDI data were compared to mass estimates obtained by PLM on a coccolith population originating from a large number of coccospheres obtained from the same cultures (colored dots in Fig. 3). The coccolith mass variability obtained by PLM was only slightly greater than that obtained by 3D-CXDI. This shows that the distribution of coccolith masses within a coccosphere is indicative of the distribution of masses within a species.

**Role of organic base plate scale (OBPS) size**. To investigate the origin of the high coccolith mass variability, a first positive correlation was found between $m$ and $p$ obeying the formula

$$m = k_p \times p^\beta \qquad (1)$$

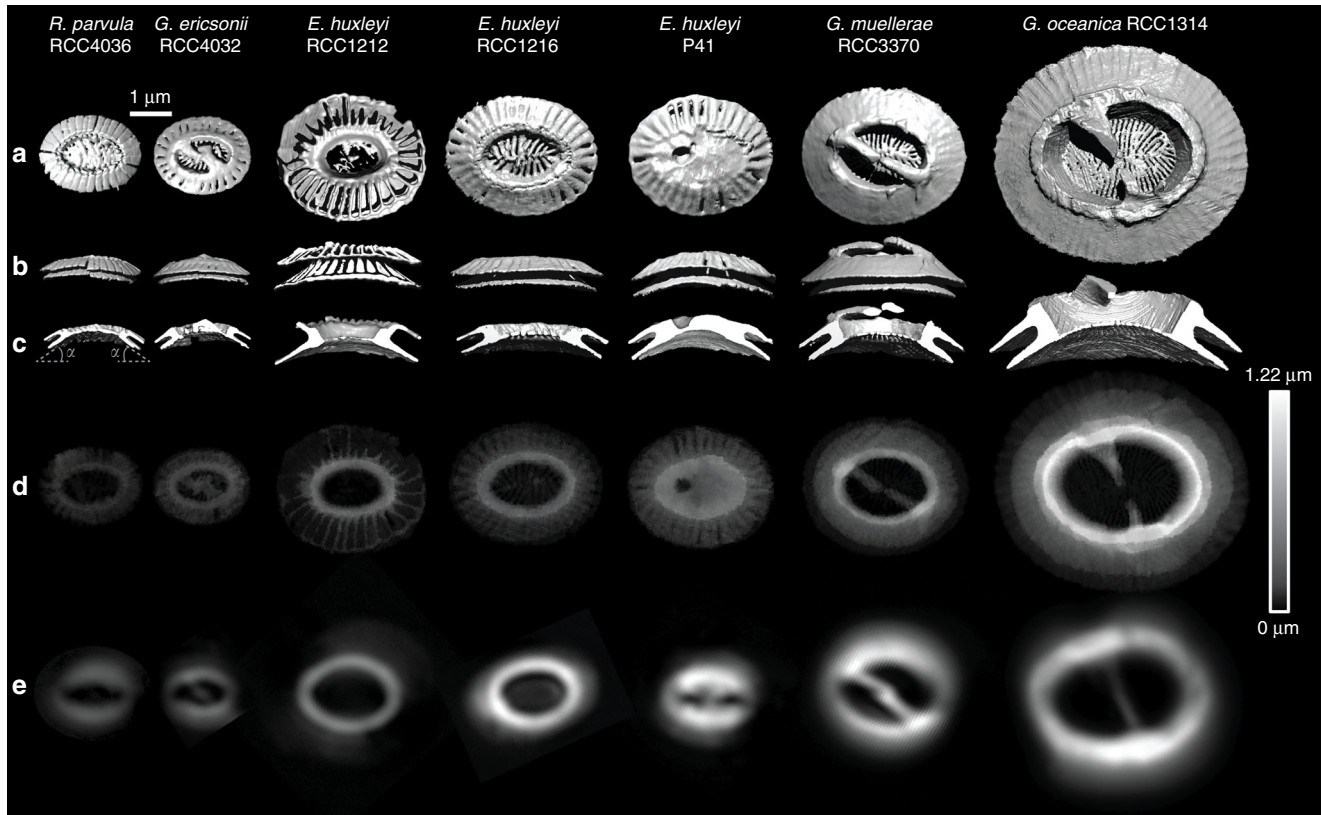

**Fig. 2** 3D-CXDI of coccoliths. **a–c** 3D-CXDI of coccoliths extracted from coccospheres and observed in: **a** distal view; **b** side view along the major axis; **c** side view along the major axis after sectioning half of the coccoliths. The angle $\alpha$ refers to the inclination of the shields. **d** In-plane thickness of the coccoliths resulting from the in-plane projection. **e** Images obtained by polarized light microscopy (PLM) displaying the distal view of coccoliths. Scale bar $= 1\,\mu m$

with $m$ the mass of a coccolith in pg, $p$ the peripheral grid perimeter of a coccolith in μm. The coefficients are $k_p = 4.92 \times 10^{-2} \pm 2.17 \times 10^{-2}$ and $\beta = 3.175 \pm 0.251$ (with 95% confidence bounds) (Fig. 4a). $k_p$ may be called the coccolith mass index. Remarkably, all studied species have the same index. Indeed, relation (1) means that the mass of a coccolith $m$ is directly linked to the perimeter $p$ of the grid, i.e., when $p$ is known the mass of the coccolith can be estimated from Eq. (1). As observed in Fig. 2d, this reflects the fact that an important part of the mass of a coccolith is located in the tube region. By looking in more detail, we observed also that the length of the proximal rim $L$ scales linearly with $p$ within species, but with a different constant of proportionality for different species (Fig. 4b). The highest $L$ over $p$ ratio is obtained for *E. huxleyi RCC1212* followed by *E. huxleyi RCC1216*, whereas *G. oceanica* is characterized by a low $L$ over $p$ ratio. Thus, CaCO$_3$ biomineralization takes place by favoring the growth of either the distal part with calcification between segments (case of *G. oceanica*) or the proximal part with longer proximal rims (case of *E. huxleyi RCC1212*). This latter correlation is undoubtedly relevant in evolutionary terms, even if the selective pressures acting on coccolith geometry are not currently well understood. We highlight also the fact that the thickness of the tube $t$ evolves almost linearly with $p$ (Fig. 4c). Thus, the exponent $\beta$ in relation (1) is close to 3 as both the length of the rim $L$ and the thickness of the tube $t$ scale more or less linearly with $p$. For the sake of clarity, the parameters $a$, $b$, $a_g$, $b_g$, $\alpha$, $p$, $L$, $t$, and $w$ are schematized in Supplementary Fig. 10. Our findings show also that $L$ and $t$ are positively correlated as reported by O'Dea et al.[46]. It is worth noting that by assuming $\beta = 3$, the parameter $k_p = 6.67 \times 10^{-2}$ ($\pm 0.4 \times 10^{-2}$ with 95% confidence bounds) is obtained. In addition, the peripheral grid

perimeter $p$ of the coccoliths scales linearly with the number $n$ of calcite segments in all measured coccoliths (Fig. 4d) as

$$p = w \times n \qquad (2)$$

with $w = 110$–$120\,nm$ corresponding to the tube average tangential width of the calcite segments at the periphery of the grid. For instance, the smallest (Fig. 2, *R. parvula*) and biggest (Fig. 2, *G. oceanica*) coccoliths display $n = 29$ and $n = 61$ segments with peripheral grid perimeters of $p = 3.32 \pm 0.18\,\mu m$ and $p = 6.92 \pm 0.19\,\mu m$ leading to a $w = 115 \pm 6\,nm$ and $w = 113 \pm 3\,nm$, respectively. Each segment of the coccoliths is composed of two types of calcite crystals, one with the $c$-axis orientation parallel to the coccolith plane and denoted R-unit ("R" for radial) and the other with the $c$-axis perpendicular to the coccolith plane (V-unit; "V" for vertical)[43,47–50]. During coccolithogenesis, the proto-coccolith ring at the periphery of the OBPS is composed of alternating V-units and R-units[43,51]. However, in *Emiliania*, *Gephyrocapsa*, and *Reticulofenestra*, mature coccoliths are mainly composed of R-units because V-units do not develop[43]. Thus, our findings lead us to propose that the periphery of the OBPS controls the mineralization site number $n$, with a R-unit nucleation site every $w$ and also a V-unit nucleation site every $w$. As V-units are not developed, the average width $w$ of the R-unit segments appears to be a constant close to 110–120 nm whatever the species. The extraction of individual segments for *R. parvula* is shown in Supplementary Fig. 11. Combining relations (1) and (2) with $w = 112\,nm$ and $\beta = 3.175$, it appears that

$$m = k_n \times n^{\beta} \qquad (3)$$

with $m$ in pg and $k_n = 4.73 \times 10^{-5}$ ($\pm 0.28 \times 10^{-5}$ with 95%

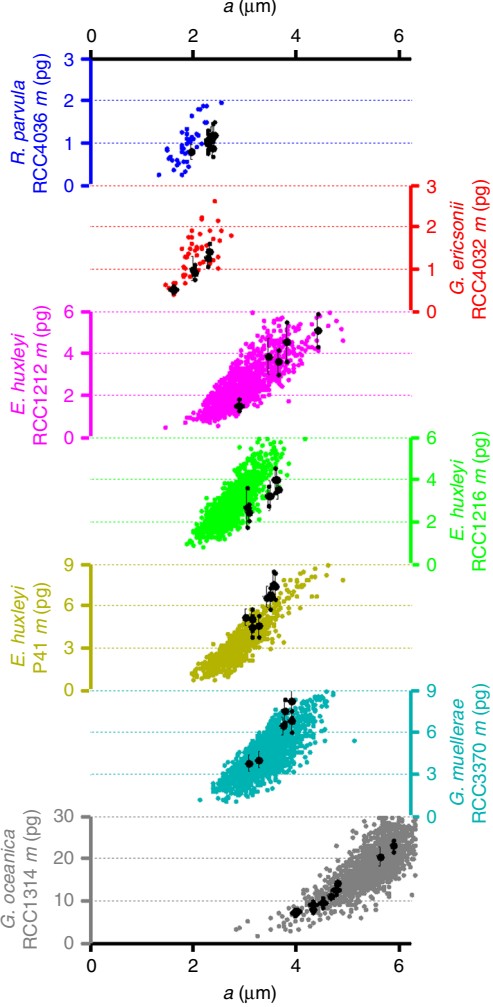

**Fig. 3** Mass of coccoliths $m$ as a function of their major axis $a$ from PLM and 3D-CXDI measurements. For 3D-CXDI (black dots), the variability includes only the intra-coccosphere variability (i.e., the coccolith mass variability within a single coccolithophore for each species). For PLM (colored dots), the variability in size and mass results from the intra- and inter-coccosphere variabilities. For the two methods coccoliths originate from the same cultures

confidence bounds) (Fig. 4e). By fixing $\beta = 3.0$, the fit to the data with relation (3) yields $k_n = 9.39 \times 10^{-5}$ ($\pm 0.66 \times 10^{-5}$ with 95% confidence bounds). This shows that the exponent $\beta$ and the prefactor $k_n$ are highly correlated in the fit. As the exponent $\beta > 1$, relation (3) means that the nucleation site number $n$ and the average mass of each segment $m/n$ are positively correlated. Among the seven explored species, *E. huxleyi* P41 is the only one for which the correlation between the number of segments and the mass of the coccolith is poor (see the circle in Fig. 4e). This can be explained by the large amount of $CaCO_3$ of the central area of the *E. huxleyi* P41 coccoliths. In this case, the mass of the grid is not small compared to the masses in the tube and shield regions. Even though strong variations in the 3D geometry of the segments are visible within species, the mass of a coccolith is thus determined at the early stage of nucleation by the number of segments and as a consequence by the size of the OBPS as illustrated in Fig. 5.

## Discussion

Our study clearly shows that the number of R-unit segments scales linearly with the perimeter length $p$ of the grid (see the

sketch in Supplementary Fig. 10 for more details), leading to an average width of the segments of 110–120 nm, whatever the species. According to the literature[38,40,43,52], nucleation of $CaCO_3$ segments occurs on the outer perimeter of the OBPS within the coccolith vesicle. We thus conclude that the $CaCO_3$ nucleation sites are at a constant spacing of 110–120 nm on the outer perimeter of the OBPS, independent of the actual size of the OBPS. An increasing or decreasing size or perimeter length of the OBPS is accomodated by the production of more or fewer segments, respectively. This in turn explains the important coccolith mass variability (with a coccolith mass ratio up to 3 between the lighter and the heavier coccoliths within a single coccosphere) by the high OBPS size variability. Our work shows also that the out-of-plane inclination of the shields of coccoliths is constant (inclined by about $\alpha \sim 30 \pm 5°$ along the major axis and $\alpha \sim 25 \pm 5°$ along the minor axis (Fig. 2b, c) compared to the coccolith plane) whatever the coccolith size. We therefore speculate that OBPS size could be determined by the cell nucleus size, which varies significantly through the cell growth/division cycle. Muller and coworkers have shown that calcification in *E. huxleyi* is largely confined to the G1 (gap 1, assimilation) cell cycle phase[21] characterized together with the S and G2 phases by a long growth period resulting from high photosynthetic activity. As the nuclear size is determined by the cytoplasmic volume rather than DNA content[53], growth of the cell during interphase may be accompanied by growth of the cell nucleus. Hence, the smallest coccoliths may be formed at the onset of the G1 phase after cell division, when the cell, the cell nucleus and the OBPS are small, whereas the largest coccoliths appear at the end of the G1 phase when the cell, the cell nucleus and the OBPS are larger. This assumption is also corroborated by 3D-CXDI results, which show clearly the correlation between the size of coccoliths and the diameter of the cell after artificially removing the mineralized part to take into account only the organic part (Supplementary Fig. 6B). Even though the positive correlation between cell size and coccolith size was already reported within and between species[44,45], further analysis using coherent X-ray diffraction imaging at cryogenic temperature on frozen-hydrated cells[54,55] or confocal microscopy on stained cells would be needed to check whether the cell nucleus size, which varies significantly through the cell growth/division cycle, could indeed regulate the size of the OBPS and therefore coccolith mass.

## Methods

**Culture**. The coccolithophore cultures were obtained from the Roscoff Culture Collection (RCC: http://www.roscoff-culture-collection.org/).

**Polarized light microscopy**. In order to measure the mass and size of detached coccoliths from cultures, we used a Leica DRM6000 light microscope with high resolution lens (Leica, HCX PL APO 100/1.47) and condenser (Leica P 1.40 Oel) and Chroma circular polarizers. The images were grabbed by high resolution cameras: a Spot Flex camera from Diagnostic Instrument (14-bit depth, 7.4-μm pixel size) for automatic coccolith selection and an ORCA Flash 4 from Hamamatsu (16-bit depth, 6.8-μm pixel size) for manual coccolith selection. Samples were prepared by settling dried coccolithophore cultures following the protocol described in ref. [28]. Detached coccoliths from *G. oceanica*, *G. muellerae*, and *E. huxleyi* were selected automatically by a deep-learning software (SYRACO)[56]. The small coccoliths of *G. ericsonii* and *R. parvula* had very dim edges that were easily merged in the background of the images and the size (length and width) of coccoliths were therefore often underestimated with the above protocol. For these species we therefore used a camera performing well in low light intensity (ORCA Flash 4) to grab coccoliths individually and manually. The size and mass measurements of the coccoliths were then performed using the protocol described in ref. [28].

**Scanning electron microscopy**. Scanning electron microscopy (SEM, LEO 1530) was performed on the same coccolithophore cells as those studied by 3D-CXDI by locating the position of each specimen. The images were obtained using the secondary electron mode of detection and with an accelerating tension of 20 kV. Specimens were previously metalized with a ~5 nm thick coat of gold. Note that

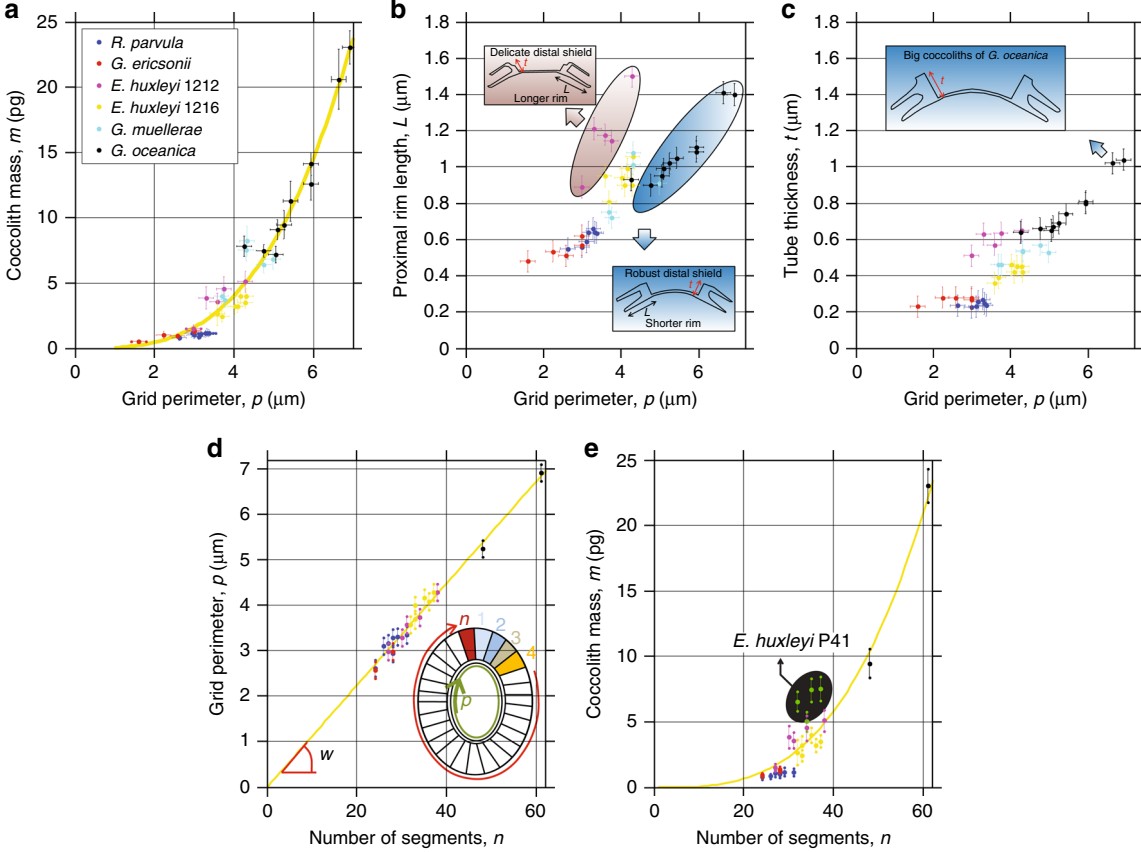

**Fig. 4** Dependence of OBPS size on the number of segments and the mass of coccoliths. **a** Mass of coccoliths $m$ as a function of the peripheral grid perimeter $p$ for all species, except $E.$ $huxleyi$ P41 for which $p$ cannot be determined because the out-of-plane thickness of the grid is comparable to that of the tube rendering impossible the identification of the frontier between the two elements. The yellow dotted line is $m = k_n \times n^\beta$ with $k_n = 4.73 \times 10^{-5}$, $\beta =$ 3.175 and $m$ in pg (coefficient of determination $R^2 = 95\%$). **b** Proximal rim length $L$ as a function of $p$. For a given $p$, coccoliths with short rims display robust distal shields (i.e., no space between the elements) while coccoliths with long rims display delicate distal shields (slit between the elements). **c** Tube thickness $t$ as a function of $p$. **d** $p$ as a function of the number of segments $n$ for coccoliths from five different coccolithophore species. The equation of the dotted line is $p = w \times n$ with $p$ in μm and $w = 0.112$ μm. The slope $w$ is equal to the average width of the tube at the periphery of the grid. For $G.$ $muellerae$, the determination of $n$ was not possible because the frontier between the successive segments is not resolved. For $E.$ $huxleyi$ P41, $p$ cannot be measured (see text above). **e** Mass of coccoliths as a function of $n$. The yellow dotted line is $m = k_n \times n^\beta$ with $k_n = 4.73 \times 10^{-5}$, $m$ in pg. By excluding the data for $E.$ $huxleyi$ P41 (shown in green) for which the central area of the coccoliths (i.e., the grid) contains a significant amount of $CaCO_3$, the coefficient of determination is $R^2 = 97\%$

3D-CXDI analysis was performed before SEM observations so that the cocco-spheres analyzed by 3D-CXDI were not metalized.

**CXDI measurements and reconstructions.** CXDI is an X-ray imaging technique well suited for visualizing at few nanometers resolution the 3D nanostructure of isolated micrometer-sized objects, e.g., biological specimens[55,57] and material science samples[30,31,58,59]. We performed the CXDI measurements at the ESRF beamline ID10[30]. The X-ray beam produced by an undulator source was mono-chromatized by a Si(111) pseudo-channel cut monochromator. Beryllium compound refractive lenses were employed to focus the beam at the sample position. The coherent fraction of the beam was finally selected by roller-blade slits opened to $10 \times 10$ μm and placed 0.55 m upstream of the sample, giving essentially a plane-wave-like illumination[60]. A sketch of the experimental setup is shown in Supplementary Fig. 1A. The sample was mounted on a horizontal ultra-precision rotation stage equipped with $x$, $y$, $z$ translations and an on-axis optical microscope. The microscopic samples were deposited on 100-nm-thick $Si_3N_4$ membranes and kept fixed by electrostatic forces. 2D diffraction patterns were recorded by a Maxipix detector having $516 \times 516$ pixels of 55 μm in size. The detector was placed 5.2 m downstream of the sample and a vacuum flight tube was used to reduce air absorption and scattering. A beamstop was inserted in front of the detector inside the vacuum flight tube to block the intense direct beam and protect the detector from radiation damage. A series of 2D diffraction patterns were recorded at sample tilt angles from $-80°(\pm5°)$ to $+80°(\pm5°)$ with $0.2°$–$0.5°$ step. The average mea-surement time per sample was 1–8 h with 4.5–25 s per frame. The 2D patterns were assembled into the 3D diffraction volume using linear interpolation. The real space image reconstruction was achieved by the iterative phase retrieval algorithm applied to the 3D diffraction volume (Supplementary Fig. 1B). Details of the phase

retrieval procedure are provided in ref. [30]. The final real space images were obtained by averaging 20 reconstructions. We used 7.0 and 8.1 keV X-rays so the real space images have the voxel size of $32.5 \times 32.5 \times 32.5$ nm$^3$ for $G.$ $oceanica$ (RCC1314), $G.$ $muellerae$ (RCC3370), $E.$ $huxleyi$ (RCC1212), $G.$ $ericsonii$ (RCC4032), and $28.8 \times 28.8 \times 28.8$ nm$^3$ for $E.$ $huxleyi$ (P41), $E.$ $huxleyi$ (RCC1216), and $R.$ $parvula$ (RCC4036), respectively. The reconstructed real space images suffer from smooth density variations due to missing data, strongly resembling the "uncon-strained modes" reported by Thibault et al.[61]. To remedy these density variations, a simple spatial flattening of the electron density was applied to the reconstructions by subtracting in real space a 3D Gaussian function centered at the mass center[31]. After subtracting the 3D Gaussian function, voxels with negative density values were set to zero. Examples of the flattening corrections are shown in Supple-mentary Fig. 1C. Chimera software was used for the visualization of the surface of the 3D volume obtained by CXDI. For each species, the tomographic slice images before the Gaussian subtraction are available in figShare (https://figshare.com/).

**PRTF and real resolution of CXDI measurements.** The real resolution of the average images was estimated using the phase retrieval transfer function (PRTF)[62]. The intersection at 0.5 threshold was used to estimate the real resolution (see for instance ref. [31]). The PRTFs were calculated for $R.$ $parvula$, $E.$ $huxleyi$ RCC1216, and $G.$ $oceanica$ (Supplementary Fig. 2). We found that the PRTFs for $R.$ $parvula$ and $E.$ $huxleyi$ RCC1216 are always higher than 0.5. From these results, we can conclude that the real resolution is close to the voxel size (voxel size $= 28.8 \times 28.8 \times 28.8$ nm$^3$). For $G.$ $oceanica$, the real resolution appears to be equal to 35.0 nm. This value is also close to the voxel size (voxel size $= 32.5 \times 32.5 \times 32.5$ nm$^3$ for the experiment done on $G.$ $oceanica$). It is worth noting that the number of scattered photons depends on the optical contrast (the same for all samples), the

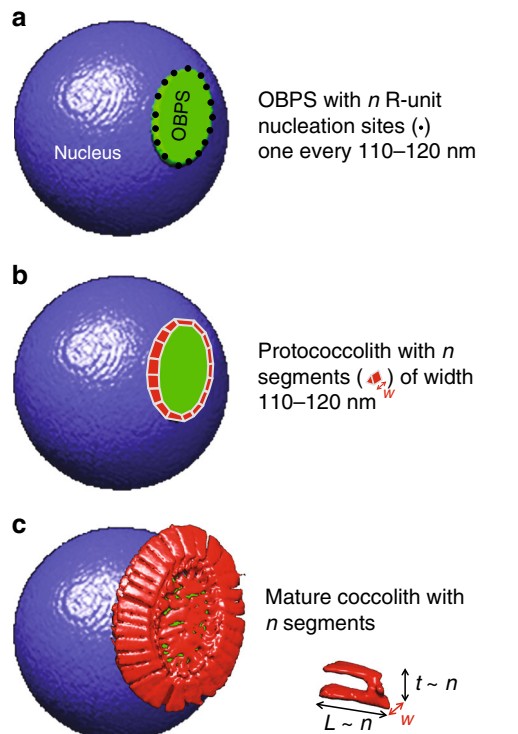

**Fig. 5** Scheme summarizing the link between OBPS size and coccolith mass. **a** Formation of the OBPS (in green) on the cell nucleus (in blue). The OBPS is decorated with $n$ R-unit nucleation sites, one every 110–120 nm in average. **b** Nucleation of the protococcolith (PC) (in red) around the OBPS. The PC contains $n$ segments with an average width $w = 110$–120 nm. **c** In-plane (forming the rim of length $L$) and out-of-plane (forming the tube of height $t$) growths of the coccolith. With $L$ and $t$ almost proportional to the segment number $n$ and with $w$ a constant equal to around 110–120 nm, the average mass of each segment is almost proportional to $n^2$. The symbol ~ refers to "almost proportional to". This leads to a coccolith mass almost proportional to $n^3$

size of the specimen (scattering volume) and the exposure time. The exposure time was chosen in order that the measured 2D diffraction patterns were fully covered by speckles up to the edge of the detector. As a result, the resolution in this study was mainly limited by the detector size.

**Segmentation of coccoliths from coccospheres**. The methodology is composed of several steps illustrated in Supplementary Fig. 3 in the case of *E. huxleyi P41*, a coccosphere having $C_N = 14$ coccoliths. In the first step, a coccolith is extracted from the whole coccosphere. The extracted coccolith is then subtracted from the coccosphere leading to a new coccosphere matrix containing $C_N - 1$ coccoliths. The methodology is repeated several times (14 times in the case of *E. huxleyi P41*) to extract all coccoliths. Coccoliths which are broken or not correctly extracted due to their close contact with neighbors were not taken into account in mass determinations.

## Data availability

Raw data generated by 3D-CXDI (i.e. reconstructed real space images before the substraction by the 3D Gaussian functions) that support the findings of this study have been deposited in https://figShare.com/account/home with the identifiers: https://doi.org/10.6084/m9.figshare.7467143.v1 for *Emiliania huxleyi RCC1212* in Supplementary Fig.7A, https://doi.org/10.6084/m9.figshare.7454078.v1 for *Emiliania huxleyi RCC1212* in Fig. 1c and Supplementary Fig.7B, https://doi.org/10.6084/m9.figshare.7454072.v1 for *Emiliania huxleyi P41* in Fig. 1c, https://doi.org/10.6084/m9.figshare.7413500.v1 for *Gephyrocapsa ericsonii RCC4032* in Fig. 1c, https://doi.org/10.6084/m9.figshare.7413494.v1 for *Gephyrocapsa muellerae RCC3370* in Fig. 1c, https://doi.org/10.6084/m9.figshare.7413491.v1 for *Gephyrocapsa oceanica RCC1314* in Fig. 1a, https://doi.org/10.6084/m9.figshare.7413377.v1 for *Reticulofenestra parvula RCC4036* in Fig. 1c. In addition, the data with the dimensions of the coccoliths are collected in Supplementary Table 1.

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

## Acknowledgements

We thank Irina Snigireva for SEM analysis. We gratefully acknowledge the European Synchrotron Radiation Facility (Grenoble, France) for provision of ID10 synchrotron beam time (EV162 and CH4936). The authors thank Jerome Kieffer and Pierre Paleo for accelerating the reconstruction code. A.G. would like to thank A.J. Pryor for fruitful discussions and advice for the use of Chimera. This software is developed by the Resource for Biocomputing, Visualization, and Informatics at the University of California, San Francisco (supported by NIGMS P41-GM103311). Funds from the Agence Nationale de la Recherche under project ANR-12-B06-0007 (CALHIS) are acknowledged.

## Author contributions

A.G. and T.B. had the idea to measure coccolithophores in three dimensions. L.B. had the idea to compare the mass obtained by 3D-CXDI with that obtained by optical microscopy. I.P. provided coccolithophore strains and culturing expertise. L.B. performed circular-polarized light microscopy. Y.C., F.Z., A.G. and T.B. managed the 3D-CXDI measurements at the synchrotron. T.B., I.P., L.B., and B.S. mounted samples on the membranes. Y.C. and T.B. performed reconstructions. A.G. and T.B. undertook analyses to segment coccoliths from the coccospheres. T.B. wrote the paper with feedback from all authors.
