## [Peer Review File · Nature Communications]

Reviewers' comments:

Reviewer #1 (Remarks to the Author):

Coccolithophores - unicellular eukaryotic phytoplankton (calcareous algae) - are the most important producers of biomass and CaCO₃ in the sea, and their metabolism is thus a major factor in global CO₂ and calcium cycles. Moreover, coccolithophores are also extremely important research subjects as indicators of local climate parameters and nutrient productivity of water columns both with respect to recent processes as well as processes that happened in earlier chapters of the history of the Earth. Precise numbers on metabolic productivity of these life forms - such as the numbers presented in the present paper - are thus important and form a major issue in current studies.

The "coccoliths" are intricate and structurally complex units of CaCO₃ exocytosed by these cells during photosynthesis.

The paper describes a very innovative and promising experimental approach to determine quantitative data on the morphology and the mass of coccoliths. The approach is based on the availability of a novel synchrotron- X-ray 3-dimensional imaging method (3D-CXDI) which obtains a space resolution of a few tens of nanometers, i.e. it is situated between light- and electron microscopy, but offers the additional advantage of supplying 3D information with comparable ease.

Using this available technique, the authors obtain a set of very valuable new information on correlation between structural details, size, shape, and mass of the coccoliths of different coccolithophore species/strands and pave the way for more very substantial studies in the field. From their new empirical correlations the authors formulate the hypothesis that the organic substrate on which the coccolith grows (the organic base plate scale) essentially determines the final shape and mass of the coccolith, as this scale controls the nucleation and supposedly the growth of the CaCO₃. The finding that the base-plate scale is the probable nucleation site of the CaCO₃ is not new and dates back at least to the 1970s. What is new is that the authors were able to "count the nucleation sites" on the base-plate. The first part of their hypothesis (i.e. about the nucleation on the base plate playing a role for final mass) is probably correct. The second part, related to "growth", is not substantiated very clearly in the manuscript, as there are many imprecise statements (see detailed comments below), which must be made more stringent. In the present data I do not see a prove of the second part of the hypothesis. Most importantly, the shown correlation between volume (corresponding to mass) and a linear size parameter (circumference) to the power of three ($V = \text{size}^3$) may well be just trivial rather than telling us something specific on coccolithophores. The volume of a truck may correlate with the size of its engine, but it is certainly not the engine which determines the volume of the truck. Given the imprecision of many statements in the present state of the manuscript I cannot not always see sufficiently well if the line of argument is both without any gaps and original. But beyond that I must raise the criticism that the final size and shape of the coccolith must also be determined by other factors. For example, some "signal" which finally stops growth (e.g. the organic membrane surrounding the coccolith elements [1]), or the size of the coccolith vesicle (the intracellular compartment in which the coccolith grows), or the growth rate of the coccolith which will depend on the transport pathways by which Ca²⁺ and HCO₃⁻ enter the coccolith vesicle [1]), etc. This at least needs to be discussed.

Given the many items of valuable information for a large community I recommend publication of the paper if the chains of argument can be made stringent. The manuscript needs some substantial revision in its text and consideration of some more literature on the subject.

[1] Yin, X., Ziegler, A., Kelm, K., Hoffmann, R., Watermeyer, P., Alexa, P., ... & Grieshaber, E. (2018). Formation and mosaicity of coccolith segment calcite of the marine algae *Emiliana huxleyi*. *Journal of phycology*, 54(1), 85-104.

DETAILED COMMENTS

Line 32 significantly impacted BY WHAT ?

Line 47-48: LITERATURE REFERENCES INSUFFICIENTLY COVERS THE EXISTING LITERATURE

55-62: WHEN DISCUSSING COCCOLITHOPHORE MASS PRODUCTIVITY ESTIMATES PERHAPS HOFFMANN'S WORK SHOULD BE REFERENCED

Hoffmann, R. et al. (2014). Insight into *Emiliana huxleyi* coccospheres by focused ion beam sectioning. *Biogeosciences Discussions*, 11(9), 12773-12797.

SHE ALSO DETERMINED THE CRYSTALLOGRAPHIC ORIENTATION OF R AND V UNITS FOR *Ehux* AS GIVEN BY THE TIME OF NUCLEATION ON THE BASE PLATE

Hoffmann, R. et al. (2014). Nanoprobe crystallographic orientation studies of isolated shield elements of the coccolithophore species *Emiliana huxleyi*. *European Journal of Mineralogy*, 26(4), 473-483.

75-77 REFERENCE MISSING TO COCCOLITH FORMATION

82 peripheral grid perimeter p . PLEASE DEFINE WHAT THIS IS SUPPOSED TO BE

108 Whereas the major axis of the coccolith varies from 2 to 6 μm : WITHIN ONE SPECIES OR BETWEEN SPECIES ?

112 which is in this case independent of the curvature of the nuclear membrane (fig. S9). PLEASE RESTATE THIS SENTENCE, E.G. "which is in this case is an experimental artefact."

113 top view thicknesses of the coccoliths resulting from integration along...
PLEASE SAY CLEARLY WHICH VARIABLE WAS INTEGRATED OVER WHICH OTHER VARIABLE

139 variations in the degree of calcification of the coccoliths produced by single coccolithophore cells. PLEASE USE MORE PRECISE WORDING. THE COCCOLITH (LITH from LITHOS) IS ALWAYS ~100% MINERAL. WHAT DO YOU MEAN HERE ?

PERHAPS YOU MEAN "individual" RATHER THAN "single" ?

142 Fig. 3 : Mass of coccoliths m as a function of their major axis a from PLM and 3D-CXDI measurements. Mass and major axis of coccoliths measured by PLM (colored dots) and 3D CXDI (black dots). PLEASE SHORTEN TO ONE SENTENCE.

151 with m the mass of a coccolith in pg , p the peripheral grid perimeter of a coccolith in μm , $k_p = 4.92 \times 10^{-2}$, $\beta = 3.175$ (Fig. 4A). WHAT ARE THE STANDARD DEVIATIONS. MASS SHOULD BE PROPORTIONAL TO VOLUME, AND VOLUME TO A LINEAR SIZE PARAMETER LIKE CIRCUMFERENCE TO THE POWER OF 3 (RATHER THAN 3.175). PLEASE DISCUSS.

153 approximation determined by the external perimeter of the grid p . This can be explained as the majority of coccolith mass for these species is located in the tube region (Figs. 2 C and D). THIS CONCLUSION APPEARS INVALID AS IT IS TRIVIAL THAT VOLUME IS PROPORTIONAL TO A LINEAR SIZE PARAMETER TO THE POWER OF THREE.

156 length of the proximal rim L scales linearly with p within species, variations take place between species. PLEASE RESTATE. THE SCALING IS STILL LINEAR FOR OTHER SPECIES, BUT WITH A DIFFERENT CONSTANT OF PROPORTIONALITY FOR DIFFERENT SPECIES.

163 We highlight also the thickness of the tube t evolves almost linearly with p in agreement with... INCOMPLETE SENTENCE

166 ... was correlated linearly ... RESTATE E.G. AS "correlates linearly"

178 with $kn = 4.73 \times 10^{-5}$ and m in pg (fig. 4E). Relation (3) means that the nucleation and the growth steps are highly and positively correlated.
PLEASE RESTATE. SAY MORE PRECISELY WHICH QUANTITIES ARE SUPPOSED TO BE CORRELATED HOW. IT IS A RATHER TRIVIAL STATEMENT THAT THE "STEPS" ARE "CORRELATED" SINCE NOTHING WILL GROW WITHOUT NUCLEATION.

179 steps are highly and positively correlated. Even though strong variations in the 3D geometry of the segments are visible within species, the mass of a coccolith is thus significantly determined at the early stage of the nucleation by the number of segments and as a consequence by the size of the obps as schemed in fig. 4F. HERE THE AUTHORS SHOULD BE MORE CAREFUL. THE OBSERVATION OF AN EMPIRICAL CORRELATION SAYS NOTHING ABOUT CAUSALITY. E.G. OBSERVING THAT OUTSIDE TEMPERATURE CORRELATES WITH THE TIME OF THE DAY DOES NOT PROVE THAT THE CLOCK DETERMINES THE TEMPERATURE.

182 ... For E. huxleyi P41, relation 3 does not stand any more as central area (i.e. the grid) of the coccoliths contains a significant amount of CaCO_3 .
PLEASE RESTATE. DO YOU WANT TO SAY THAT EQUATION 3 IS INVALID FOR EHUX WHILE IT APPLIES FOR THE TWO OTHER SPECIES ? IF SO, THE SECOND PART OF THE SENTENCE SHOULD DESCRIBE THE DIFFERENCE BETWEEN THE SPECIES, AND NOT JUST A PROPERTY OF EHUX. "A SIGNIFICANT AMOUNT OF CaCO_3 " IS CONTAINED IN ANYTHING MADE OF CALCITE !!!!

187 Evolution... PLEASE USE ANOTHER TERM FOR THIS THROUGHOUT THE TEXT

188 species, except E. huxleyi P41 for which p can't be determined. WHY ?

188 species, except E. huxleyi P41 for which p can't be determined. The yellow dotted line is $m = kp \times p^\beta$ with m in pg, p in μm , $kp = 4.92 \times 10^{-2}$, $\beta = 3.175$ (coefficient of determination $R^2 = 95\%$). PLEASE REPORT STANDARD DEVIATIONS. IS THE DEVIATION FROM 3 OF THE POWER PARAMETER SIGNIFICANT ?

190, 192, 193, 197 ... Evolution ... PLEASE USE ANOTHER TERM

191 given p, coccoliths with short rims display robust distal shields; coccoliths with long rims display delicate distal shields. PLEASE DEFINE THE TERMS "robust" AND "delicate" IN A

QUANTITATIVE WAY.

196 tube at the periphery of the grid. For *G. muelleriae* and *E. huxleyi* P41, it was not possible to determine n and p , respectively. WHY ?

198 segment number n . The yellow dotted line is $m = kn \times n\beta$ with $kn = 4.73 \times 10^{-5}$, m in μg . WHAT ABOUT BETA ? PLEASE GIVE STANDARD DEVIATIONS FOR ALL PARAMETERS.

200 ... contains significant amount of CaCO_3 SEE COMMENT Line #182

201 ... (F) Scheme... THIS SCHEME ILLUSTRATES WHY THE SIZE OF THE INNER RING IS DETERMINED BY THE BASE PLATE. THIS IS NOT A NEW IDEA, AS YOU WILL ADMIT. BUT THIS SCHEME IN NO WAY ILLUSTRATES WHY THE FINAL (!) SIZE SHOULD CORRELATE WITH THE INITIAL SIZE.

203 unit nucleation sites, one every 112 nm in average I UNDERSTOOD YOU SAID THIS DEPENDS ON SPECIES ! ?

206 The bidimensional growth is proportional to the segment number n . PLEASE BE PRECISE ! HOW CAN "GROWTH" BE PROPORTIONAL TO AN INTEGER "NUMBER" ? IS IT GROWTH SPEED ? GROWTH RATE ? DOES THE RATE REALLY JUST JUMP IN STEPS OF INTEGER NUMBERS ?

210 conditions, pH, salinity, carbonate chemistry to forecast the coccolithophorid biomineralization upon climatic changes PLEASE RESTATE ! WHAT WOULD BE THE MEANING OF "biomineralization upon changes" ?

212 nucleation and the growth of a protococcolith and as a consequence the mass of a mature coccolith depends strongly on the obps size PLEASE RESTATE. NUCLEATION DOES NOT DEPEND ON THE obps SIZE. IT MAY BE THAT THE NUMBER OF NUCLEI DEPENDS ON THE obps SIZE. AGAIN, WHAT PRECISELY IS MEANT BY "GROWTH". GROWTH, AGAIN, DOES NOT DEPEND ON THE obps SIZE, I WOULD BET IT DEPENDS ON SUPERSATURATION AND MATERIAL TRANSPORT. IT MAY BE THAT THE FINAL MASS OF THE COCCOLITH DEPENDS ON THE ORIGINAL obps SIZE, BUT NOT "GROWTH". THE INACCURACY OF THESE STATEMENTS MAKES THE PAPER RATHER CONFUSING.

214 within a coccosphere appears to be related to the obps size variability. THE "appears" MAKES THIS A RATHER VAGUE STATEMENT. IF THIS PORPOSITION IS CORRECT AT ALL, THE LOGICAL REASONING BEHIND IT NEEDS TO BE DESCRIBED.

230 our new obps size - coccolith mass framework established the size of obps (and therefore coccolith mass) is presumably largely regulated by the cell size and the cell nucleus size, I DO NOT UNDERSTAND THIS SENTENCE. WHAT IS A "size-coccolith" ? WHAT IS A "mass framework" ? TO WHICH SUBJECT DOES THE PREDICATE "is" REFER ? ETC..

1 The dominant role of the organic base plate scale in the mass of coccoliths revealed by X-ray tomography. THIS TITLE SAYS LITERALLY THAT THE (MASS) OF THE BASE PLATE SCALE DOMINATES THE MASS OF THE COCCOLITHS. HOWEVER, THE AUTHORS DO NOT APPEAR TO MEAN THIS. OR ? "DOMINATION" IS NOT A VERY APPROPRIATE TERM HERE ANYWAY, UNLESS OTHER FACTORS DETERMINING THE MASS OF THE COCCOLITHS WERE MEASURED AND DISCUSSED. ONLY THEN THE AUTHORS WOULD BE ABLE TO SHOW THAT THE SIZE(!?) OF THE BASE PLATE SCALE DOMINATES THE OTHER FACTORS.

19 with a resolution close to 30 nm. After isolating... PLEASE EXPRESS THIS MORE PRECISELY. DID YOU ISOLATE COCCOLITHS IN THE IMAGES OR DID YOU ISOLATE THEM BY "MECHANICAL" PREPARATION

21 perimeter. Assuming that this area is reminiscent of the organic base plate scale... PLEASE SUBSTANTIATE THIS ASSUMPTION AND REPLACE "reminiscent" BY A MORE APPROPRIATE WORD

23 CaCO₃ nucleation site number and the growth of coccoliths... PERHAPS "number of nucleation sites" WOULD BE EASIER TO UNDERSTAND

23 CaCO₃ nucleation site number and the growth of coccoliths, strengthening the interplay between photosynthesis and biomineralization. PLEASE RESTATE: WHO STRENGTHENS WHAT ? THE COCCOLITHS CERTAINLY DO NOT STRENGTHEN THIS INTERPLAY.

Reviewer #2 (Remarks to the Author):

I read with a lot of attention the manuscript presented by Beuvier et al., entitled "The dominant role of the organic base plate scale in the mass of coccoliths revealed by X-ray tomography". This work presents a careful analysis of the relationships between the sizes of several specific structural features of a coccolith and between the coccolith structural features and its organic base plate scale. It evidences the role of the cell and cell nucleus sizes in the biomineralization process of these marine species. The conclusions are robust, based on an extended set of data, including several coccolith species and corresponding to a considerable amount of experimental results. The supplementary data are very helpful in order to follow all the steps of the analysis process, obviously performed in a very careful and systematic way. The whole article is very well written and should be readable by a large audience.

In addition, it makes no doubt that these original findings could only be obtained by the use of 3D x-ray CDI, which provides in a non destructive way, a 3D representation of the individual coccolith volumes, while minimizing the manipulation of these fragile biominerals. Neither optical nor electron microscopy nor x-ray scanning diffraction could provide equivalent results.

For all these reasons, I think that this manuscript is suitable for Nature Communications, provided that the authors consider the few questions, reported below, in order to improve the strength of the manuscript.

1 – Resolution estimation: the resolution estimation is presented in the Supplementary material for *E. Huxleyi*, one of the largest of the analyzed coccoliths. First, I think that the shown PRTF is integrated over all directions, while I think that the effective resolution in one direction should rather be presented. Second, I would like to see the resolution estimation for all species, including

the smaller one, where one may expect less scattered photons (unless the acquisition time has been scaled accordingly).

2 – line 68, the typical size of the coccoliths should be given, as the argument directly refers to the resolution capability of 3DCDI.

3 – Please unify the terminology associated to the method: it is sometimes referred to as 3DCDI or to x-ray tomography. I think 3D CDI is more appropriate.

4 – Supplementary material: line 83 “blue curve” should be “purple curve”

Reviewer #3 (Remarks to the Author):

The manuscript by Beuvier et al employs a novel approach to determining the mass of coccoliths of three ecologically and evolutionarily important coccolithophore species. The novel application of X-ray tomography allows accurate determination of the volume of individual coccoliths along with detailed 3-D structure, allowing the relative mass proportions of different regions of the coccolith to be measured. The authors have provided a convincing validity analysis of the approach and the data that is presented is also convincing.

The main conclusions are: 1). The size and curvature of the nucleus determines the size and curvature of the coccoliths, and it is proposed that variations in the size of the nucleus with cell cycle account for the observed coccolith size distributions. This largely confirms what previous studies have also proposed. 2). The mass of the coccolith is determined by the number of coccolith segments in the central tube area perimeter. It is concluded that this in turn is determined by the number of crystal nucleation sites provided by an organic base plate scale.

While the data are convincing, I think that the manuscript could have been strengthened by the provision of additional data to support the conclusions. Direct determination of nuclear size would have allowed a more robust relationship between nucleus and coccolith to be made. It should not be difficult to obtain such measurements.

The conclusion that the obps structure underlies this structure is postulated in a number of other publications. However, very little is known about the nucleating so-called baseplate structures in coccolithophores, particularly the species under study. The conclusions are, therefore somewhat speculative and I am not sure if the conclusion implicit in the title “The dominant role of the organic base plate scale in the mass of coccoliths revealed by X-ray tomography” is fully supported by the data provided since no information on the actual obps is provided.

This letter addresses the referees' comments in a point-by-point manner.

The reviewers' comments are in black and our answers are in blue.

Reviewers' comments:

Reviewer #1 (Remarks to the Author):

Coccolithophores - unicellular eukaryotic phytoplankton (calcareous algae) - are the most important producers of biomass and CaCO₃ in the sea, and their metabolism is thus a major factor in global CO₂ and calcium cycles. Moreover, coccolithophores are also extremely important research subjects as indicators of local climate parameters and nutrient productivity of water columns both with respect to recent processes as well as processes that happened in earlier chapters of the history of the Earth. Precise numbers on metabolic productivity of these life forms - such as the numbers presented in the present paper - are thus important and form a major issue in current studies.

The "coccoliths" are intricate and structurally complex units of CaCO₃ exocytosed by these cells during photosynthesis.

The paper describes a very innovative and promising experimental approach to determine quantitative data on the morphology and the mass of coccoliths. The approach is based on the availability of a novel synchrotron- X-ray 3-dimensional imaging method (3D-CXDI) which obtains a space resolution of a few tens of nanometers, i.e. it is situated between light- and electron microscopy, but offers the additional advantage of supplying 3D information with comparable ease.

Using this available technique, the authors obtain a set of very valuable new information on correlation between structural details, size, shape, and mass of the coccoliths of different coccolithophore species/strands and pave the way for more very substantial studies in the field. From their new empirical correlations the authors formulate the hypothesis that the organic substrate on which the coccolith grows (the organic base plate scale) essentially determines the final shape and mass of the coccolith, as this scale controls the nucleation and supposedly the growth of the CaCO₃. The finding that the base-plate scale is the probable nucleation site of the CaCO₃ is not new and dates back at least to the 1970s. What is new is that the authors were able to "count the nucleation sites" on the base-plate. The first part of their hypothesis (i.e. about the nucleation on the base plate playing a role for final mass) is probably correct.

The second part, related to "growth", is not substantiated very clearly in the manuscript, as there are many imprecise statements (see detailed comments below), which must be made more stringent. In the present data I do not see a prove of the second part of the hypothesis.

We believe that the second part of our results (i.e. about the growth of the coccolith) was clarified. In particular, as explained deeply later, the term "growth" was replaced by measurable quantities such as the "size" or the "mass".

Most importantly, the shown correlation between volume (corresponding to mass) and a linear size parameter (circumference) to the power of three ($V = \text{size}^3$) may well be just trivial rather than telling us something specific on coccolithophores. The volume of a truck may correlate with the size of its engine, but it is certainly not the engine which determines the volume of the truck. Given the imprecision of many statements in the present state of the manuscript I cannot not always see sufficiently well if the line of argument is both without any gaps and original.

Our results shows that the volume V or the mass m are almost proportional to the cube of the grid perimeter p ($m \sim p^3$; the symbol " \sim " means here "proportional to"). This relationship seems relatively trivial. It refers to the homothetic growth in the 3 directions of space.

However, where the article becomes interesting and not trivial is when one takes into account the linear dependence between p and n .

Let us take the example of the human skeleton. It is made up of 206 bones in adulthood.

Whatever the total bone volume of a person, the number of bones is unchanged. It is constant and equal to 206. A person with low bone volume has as many bones as a person with high bone volume. However, there is a link between a person's bone volume and the average mass of each bone. In other words, on average, a person with high bone volume will have larger bones on average. For example, measurements of femur size and abdominal circumference can be used to estimate the total mass of a fetus [see Frank P. Hadlock et al, Estimation of fetal weight with the use of head, body, and femur measurements-A prospective study, February 1, 1985 Volume 151, Issue 3, Pages 333-337]. However, the mass of a fetus is not an affine function of the cube the length of the femur. Similarly, the mass m of an adult is not proportional to the cube of the adult's size t_a but rather evolves according to $m \sim t_a^2$ (the symbol " \sim " means here "proportional to"). See body mass index. In the case of coccoliths, the number of segments is not constant. There is a relationship between the mass of the coccoliths m and the number of segments n . The greater the mass, the greater the number of segments ($m \sim n^\beta$ with $\beta = 3$). In addition, the higher the number of segments, the higher the mass of each segment. This originates from the fact that $\beta > 2$.

Line 178, we replace the following sentence :

"Relation (3) means that the nucleation and the growth steps are highly and positively correlated."

by

" As the exponent $\beta > 2$, relation (3) means that the nucleation site number n and the average mass of each segment m/n are positively correlated."

Thus, the 3 key quantities of each segment, width w , height t and length L , have developments which are not trivial. w is constant and close to 112 nm whatever the species and mass of the coccolithe. This parameter is therefore a constant. Conversely, t and L evolve almost linearly with the number of segments. Thus the mass of each segment is not defined by its width w , but by 2 parameters which are t and L and which are both proportional to n . We believe this point is correctly described in the article.

But beyond that I must raise the criticism that the final size and shape of the coccolith must also be determined by other factors. For example, some "signal" which finally stops growth (e.g. the organic membrane surrounding the coccolith elements [1]), or the size of the coccolith vesicle (the intracellular compartment in which the coccolith grows), or the growth rate of the coccolith which will depend on the transport pathways by which Ca^{2+} and HCO_3^- enter the coccolith vesicle [1]), etc. This at least needs to be discussed.

Of course the factors mentioned play important role in coccolithogenesis in general. However, our results do not provide additional information on growth mechanisms in terms of ion transport and growth rates. It is therefore difficult to discuss this in the article. Our study

shows that the final size and the mass of a coccolith are directly related to the grid perimeter. Because the inclination angles of the segments along major and minor axes are constant independent on the coccolith size we argue that the grid perimeter is given by the size of the cell nucleus. To our opinion this provides original explanation of size variability of coccoliths observed on a single coccusphere.

Given the many items of valuable information for a large community I recommend publication of the paper if the chains of argument can be made stringent. The manuscript needs some substantial revision in its text and consideration of some more literature on the subject.

[1] Yin, X., Ziegler, A., Kelm, K., Hoffmann, R., Watermeyer, P., Alexa, P., ... & Griesshaber, E. (2018). Formation and mosaicity of coccolith segment calcite of the marine algae *Emiliania huxleyi*. *Journal of phycology*, 54(1), 85-104.

We thank reviewer for drawing our attention to the paper of Yin that we were not aware of on the time of submission. We agree to cite the article because it contains several interesting TEM images which show that the coccolith vesicle is intercalated between the reticular body and the nucleus for *E. huxleyi*. As a consequence, we add this reference in the following text :

Line 77-79 : " The proximal side of the *cv* is closely apposed to the nuclear membrane and the distal side is intimately associated with a reticular body (*rb*) in *E. huxleyi* and *G. oceanica*^{35-36-newred} and probably also in *R. parvula*. "

In addition, as the authors observed that 2 coccoliths in formation can be found inside the cell, we replaced line 76 " the coccoliths are not in direct contact with seawater but are rather formed inside the cell, one at a time " by " the coccoliths are not in direct contact with seawater but are rather formed inside the cell, **generally** one at a time "

Interestingly, the authors reported also that, in some rare case (4%), immature coccoliths are formed when the proximal membrane of the coccolith vesicle (*cv*) is not in contact with the nuclear envelope (*ne*). They highlighted the important role of the close contact between the *cv* and the *ne* in the growth of the coccolith and proposed that the *ne* contains high Ca^{2+} concentration whereas the reticular body (*rb*) contains the HCO_3^- ions. The growth of coccoliths sandwiched between the *ne* and the *rb* takes place by combining Ca^{2+} and HCO_3^- . In our study, we were not able to reveal the presence of the organelles, neither the *rb* nor the nucleus. Thus, it is not possible for us to correlate these assumptions with our observations.

DETAILED COMMENTS

Line 32 significantly impacted BY WHAT ?

We think that the sentence is grammatically correct that is the algae have impacted global biogeochemical cycles and not other way around.

The objective of this sentence is to stress that the metabolism of the coccolithophores have impacted their environment (the ocean) by absorbing CO_2 and sequestering CaCO_3 . We do not believe that this sentence has to be changed.

Line 47-48: LITERATURE REFERENCES INSUFFICIENTLY COVERS THE EXISTING

LITERATURE

One reference from Hoegh-Guldberg, Coral Reefs Under Rapid Climate Change and Ocean Acidification, SCIENCE VOL 318, 14 DECEMBER 2007 was added Line 44 to support the ocean acidification. The reference 6 was placed also in Line 44. These 2 references are sufficient to highlight the importance of CO₂ in the ocean chemistry and ocean acidification. In addition, the following 2 sentences

" This global ocean change affects a variety of marine life forms, in particular calcifying organisms ⁶. Ocean acidification has been reported to decrease the degree of biogenic calcification of corals ⁷⁻⁸, echinoderms ⁹ and foraminifera ¹⁰⁻¹¹. "

were replaced by :

" This global ocean change affects a variety of marine life forms. In particular ocean acidification has been reported to decrease the degree of biogenic calcification of corals ⁷⁻⁸, echinoderms ⁹ and foraminifera ¹⁰⁻¹¹. "

55-62: WHEN DISCUSSING COCCOLITHOPHORE MASS PRODUCTIVITY ESTIMATES PERHAPS HOFFMANN'S WORK SHOULD BE REFERENCED

Hoffmann, R. et al. (2014). Insight into *Emiliania huxleyi* coccospheres by focused ion beam sectioning. Biogeosciences Discussions, 11(9), 12773-12797.

Lines 55-62, the available 3 techniques used to estimate the mass of coccolith are already presented :

- in-plane and out-of-plane 2D images (ref17) (note that to be clearer, we replaced line 57-58 "Mass can be estimated from in-plane and out-of-plane 2D images" by "Mass can be estimated from in-plane and out-of-plane 2D SEM or optical microscopy images"). This article provides the shape factor of different species of coccoliths. This shape factor, noted k_s , correlates the mass of a coccolith with its length L (L is the major axis of the coccolith) : $m = k_s \times L^3 \times \rho$. Here, ρ is the density of coccolith $\rho = 2.7 \text{ g/cm}^3$.

- resonance frequency difference (ref23)

- polarized light microscopy (ref 16, 24, 25, 26, 27)

The reference of Hoffman, R. et al (2014) used FIB-SEM. This technique allows seeing inside coccospheres by sectioning them. The use of FIB-SEM is thus interesting to count the coccoliths of coccospheres containing several layers of coccoliths as in the case of *E. huxleyi*. In the article of Hoffman, the calculation of the mass of the coccosphere was done by counting the number of coccoliths and by assuming that the mass of the coccolith is a cubic function of the length of the coccolith $m = k_s \times L^3 \times \rho$. They were able to determine the number of coccoliths per cell and then they used the density, the length and the shape factors of ref 17 to extract the mass of the coccosphere. In our opinion, the paper of Hoffman does not provide a new method to measure the mass of coccolith. We prefer not to cite this article in this paragraph related to the methods of mass measurements.

SHE ALSO DETERMINED THE CRYSTALLOGRAPHIC ORIENTATION OF R AND V

UNITS FOR Ehux AS GIVEN BY THE TIME OF NUCLEATION ON THE BASE PLATE

Hoffmann, R. et al. (2014). Nanoprobe crystallographic orientation studies of isolated shield elements of the coccolithophore species *Emiliana huxleyi*. *European Journal of Mineralogy*, 26(4), 473-483.

Thank you for drawing our attention to this paper. We introduce this reference because it confirms by transmission electron microscopy the orientation of the R-unit and V-unit calcite crystals. The interesting point of this paper is the information on the V-unit crystals. We have also to detail the history of the study of the crystalline structure of coccoliths from Watanabe (1967) to Hoffman (2014). For this purpose, line 172, we have added a sentence to explain deeper the V/R model of the segments and to cite the papers which helps to understand the orientation of the V-unit and R-unit crystals by TEM :

"Each segment of the coccoliths is composed of 2 types of calcite crystals, one with the c-axis orientation parallel to the coccolith plane and denoted R-unit ("R" for radial) and the other with the c-axis perpendicular to the coccolith plane (V-unit; "V" for vertical).[ref Watanabe 1967; S. Mann and N. H. C. Sparks 1988; Young 1992; Saruwatari 2008; Hoffman 2014]. Even if the initial crystals of the proto-coccoliths are deposited with alternating radial and vertical c-axis orientation (the V/R nucleation model) [Young 1992; Didymus 1994;], the volume of mature coccoliths is mainly composed of R-unit crystals in *Emiliana* and *Gephyrocapsa* families due to the overgrowth of the R-units crystals."

Line 173, we replace "These findings" by "Our findings"

In addition, line 167, we remove "(R_unit crystals)" because the crystallographic orientation of the segments is now explained line 172.

75-77 REFERENCE MISSING TO COCCOLITH FORMATION

2 recent references are already available line 79. To complete the references, line 77 we cite 2 old papers, the one of Wilbur and Watabe in 1963 and the one of van der Wal in Protoplasma 1983 which gives evidence of the presence of the coccolith vesicle between the cell nucleus and the reticulum body.

82 peripheral grid perimeter p . PLEASE DEFINE WHAT THIS IS SUPPOSED TO BE

Line 82, we replace :

"In other words, the perimeter of the *obps* may be similar to the CaCO_3 peripheral grid perimeter p "

by

"In other words, the perimeter of the *obps* may be similar to the one of the peripheral grid, which is constructed by the intersection between the tube and the grid (called also "central area). By assuming that the peripheral grid perimeter is an ellipse, the measurement of the semi-major axis a and the semi-minor axis b of the grid allows to determine p from $p = \pi\sqrt{2(a^2 + b^2) - 0.5(a - b)^2}$."

In addition, line 83, we replace "central area (called also "grid") size" by "peripheral grid perimeter"

108 Whereas the major axis of the coccolith varies from 2 to 6 μm : WITHIN ONE SPECIES OR BETWEEN SPECIES ?

Line 108, we added "between species"

112 which is in this case independent of the curvature of the nuclear membrane (fig. S9). PLEASE RESTATE THIS SENTENCE, E.G. "which is in this case is an experimental artefact."

Line 112, we agree with the reviewer.

113 top view thicknesses of the coccoliths resulting from integration along... PLEASE SAY CLEARLY WHICH VARIABLE WAS INTEGRATED OVER WHICH OTHER VARIABLE

Line 113, we replace the following sentence :

"The top view thicknesses of the coccoliths resulting from integration along the direction normal to the plane of the coccolith are shown in Fig. 2D."

by

"The top view thicknesses of the coccoliths are shown in Fig. 2D. They are calculated by multiplying the cubic root of one voxel (i.e. 28.8 nm or 32.5 nm) by the number of voxels along the direction normal to the plane of the coccolith having an intensity greater than the isovalue. The isovalue is the voxel intensity below which a voxel is considered as empty and above which it is considered as filled by calcium carbonate."

139 variations in the degree of calcification of the coccoliths produced by single coccolithophore cells. PLEASE USE MORE PRECISE WORDING. THE COCCOLITH (LITH from LITHOS) IS ALWAYS ~100% MINERAL. WHAT DO YOU MEAN HERE ?

We agree with the reviewer. We replace the following sentence :

" Coccolith mass variability was greater in PLM (variability between a population of coccospheres from the same culture) than in 3D-CXDI (variation on individual coccospheres), but the results show that a large part of the variability in coccolith mass within a population of cells can undoubtedly be attributed to variations in the degree of calcification of the coccoliths produced by single coccolithophore cells."

by

"Amazingly, the coccolith mass variability obtained by PLM was only slightly greater than the one obtained by 3D-CXDI. This shows that a large part of this variability comes from variability between a population of coccoliths coming from individual coccospheres."

PERHAPS YOU MEAN "individual" RATHER THAN "single" ?

We agree that "individual" is more appropriate than "single". Lines 126, 130 and 147, we let "single".

142 Fig. 3 : Mass of coccoliths m as a function of their major axis a from PLM and 3D-CXDI measurements. Mass and major axis of coccoliths measured by PLM (colored dots) and 3D CXDI (black dots). PLEASE SHORTEN TO ONE SENTENCE.

We shorten the two previous sentences by the following sentence :
" Mass of coccoliths m as a function of their major axis a from 3D-CXDI and PLM measurements."

and the order of the next sentences was modified as :

" For 3D-CXDI (black dots), the variability includes only the intra-coccosphere variability (i.e. the coccolith mass variability within a single coccolithophore for each species). For PLM (colored dots), the variability in size and mass results from the intra- and inter-coccosphere variabilities. For the two methods coccoliths originate from the same cultures."

151 with m the mass of a coccolith in pg, p the peripheral grid perimeter of a coccolith in μm , $k_p = 4.92 \times 10^{-2}$, $\beta = 3.175$ (Fig. 4A). WHAT ARE THE STANDARD DEVIATIONS. MASS SHOULD BE PROPORTIONAL TO VOLUME, AND VOLUME TO A LINEAR SIZE PARAMETER LIKE CIRCUMFERENCE TO THE POWER OF 3 (RATHER THAN 3.175). PLEASE DISCUSS.

Line 151, we replace :

"with m the mass of a coccolith in pg, p the peripheral grid perimeter of a coccolith in μm , $k_p = 4.92 \times 10^{-2}$, $\beta = 3.175$ "

by

"with m the mass of a coccolith in pg, p the peripheral grid perimeter of a coccolith in μm . The coefficients are $k_p = 0.04925 \pm 0.02170$ and $\beta = 3.175 \pm 0.251$ (with 95% confidence bounds)."

Line 176, we modify also

"relations (1) and (2) with $w = 112 \text{ nm}$, "

by

"relations (1) and (2) with $w = 112 \text{ nm}$ and $\beta = 3.175$,"

Now, what is the difference between $\beta = 3.175$ and $\beta = 3.0$ in the relation $m = k_p \times p^\beta$? The 2 curves, one with $\beta = 3.175$ and another one with $\beta = 3.0$ are plotted below in A and B, respectively :

The 2 curves are close to each other.

However, the first one (with $\beta = 3.175$) was obtained by keeping free the variable during the fit whereas the second one (with $\beta = 3.0$) was obtained by assuming/fixing this parameter.

Without a priori, we prefer to let $\beta = 3.175$ (mean value of β) even if the quite high standard deviation of β lets the possibility that $\beta = 3$. Line 165, the following sentence is added :

"It is worth noting that by assuming $\beta = 3$, the parameter $k_p = 6.67 \times 10^{-2}$ ($\pm 0.4 \times 10^{-2}$ with 95% confidence bounds) is obtained. "

In addition, line 178, the sentence :

"with $k_n = 4.73 \times 10^{-5}$ and m in pg (fig. 4E)."

is replaced by

" with m in pg and $k_n = 4.73 \times 10^{-5}$ ($\pm 0.28 \times 10^{-5}$ with 95% confidence bounds) for $\beta = 3.175$ (fig. 4E). By fixing $\beta = 3.0$, the fit to the data with relation 3 yields $k_n = 9.39 \times 10^{-5}$ ($\pm 0.66 \times 10^{-5}$ with 95% confidence bounds). This shows that a small change in the exponent β has a big effect on k_n ."

Line 198, we replace :

" The yellow dotted line is $m = k_n \times n^\beta$ with $k_n = 4.73 \times 10^{-5}$, m in pg."

by

" The yellow dotted line is $m = k_n \times n^\beta$ with $k_n = 4.73 \times 10^{-5}$, $\beta = 3.175$ and m in pg."

153 approximation determined by the external perimeter of the grid p . This can be explained as the majority of coccolith mass for these species is located in the tube region (Figs. 2 C and D). THIS CONCLUSION APPEARS INVALID AS IT IS TRIVIAL THAT VOLUME IS

PROPORTIONAL TO A LINEAR SIZE PARAMETER TO THE POWER OF THREE.

Lines 115-116, we have already said that "...the thickness is maximal in the tube region of coccoliths.". Thus it is not necessary to repeat that "the majority of coccolith mass for these species is located in the tube region". The following sentence is deleted :

" This can be explained as the majority of coccolith mass for these species is located in the tube region (Figs. 2 C and D)."

156 length of the proximal rim L scales linearly with p within species, variations take place between species. PLEASE RESTATE. THE SCALING IS STILL LINEAR FOR OTHER SPECIES, BUT WITH A DIFFERENT CONSTANT OF PROPORTIONALITY FOR DIFFERENT SPECIES.

The sentence is restated :

" The length of the proximal rim L scales linearly with p within species, but with a different constant of proportionality for different species."

163 We highlight also the thickness of the tube t evolves almost linearly with p in agreement with... INCOMPLETE SENTENCE

Lines 163-164, the following sentence :

"We highlight also the thickness of the tube t evolves almost linearly with p in agreement with literature ³⁹ (Fig. 4C)."

was replaced by :

"We highlight also the thickness of the tube t evolves almost linearly with p (Fig. 4C)."

and we add one sentence line 165 (and before the new sentence "It is worth noting ..." :

"Our findings show also that L and t are positively correlated as reported by O'Dea et al ³⁹"

166 ... was correlated linearly ... RESTATE E.G. AS "correlates linearly"

Because the previous sentence has already the word "correlated", we prefer to restate the sentence as :

"scales linearly"

178 with $kn = 4.73 \times 10^{-5}$ and m in μg (fig. 4E). Relation (3) means that the nucleation and the growth steps are highly and positively correlated.

PLEASE RESTATE. SAY MORE PRECISELY WHICH QUANTITIES ARE SUPPOSED TO BE CORRELATED HOW. IT IS A RATHER TRIVIAL STATEMENT THAT THE "STEPS" ARE "CORRELATED" SINCE NOTHING WILL GROW WITHOUT NUCLEATION.

Line 178, we replace the following sentence :

"Relation (3) means that the nucleation and the growth steps are highly and positively correlated."

by

" As the exponent $\beta > 2$, relation (3) means that the nucleation site number n and the average mass of each segment m/n are positively correlated."

179 steps are highly and positively correlated. Even though strong variations in the 3D geometry of the segments are visible within species, the mass of a coccolith is thus significantly determined at the early stage of the nucleation by the number of segments and as a consequence by the size of the obps as schemed in fig. 4F. HERE THE AUTHORS SHOULD BE MORE CAREFUL. THE OBSERVATION OF AN EMPIRICAL CORRELATION SAYS NOTHING ABOUT CAUSALITY. E.G. OBSERVING THAT OUTSIDE TEMPERATURE CORRELATES WITH THE TIME OF THE DAY DOES NOT PROVE THAT THE CLOCK DETERMINES THE TEMPERATURE.

In order to take into account this remark, we replace "significantly determined" by "scheduled". The coccolithogenesis begins with the formation of the obps. If this obps is small, the mass of the mature coccolith after nucleation and growth will be small. If the obps is big, the mass of the mature coccolith will be big. There is a chronology in the events of the coccolithogenesis. It begins with the formation of the obps, continue with the CaCO_3 nucleation and end up with the growth of the coccolith. Even if there is no causality between obps size and coccolith mass, the chronology combined with the relation (3) shows that the mass of a coccolith is scheduled as soon as the obps is formed.

182 ... For *E. huxleyi* P41, relation 3 does not stand any more as central area (i.e. the grid) of the coccoliths contains a significant amount of CaCO_3 .

PLEASE RESTATE. DO YOU WANT TO SAY THAT EQUATION 3 IS INVALID FOR EHUX WHILE IT APPLIES FOR THE TWO OTHER SPECIES ? IF SO, THE SECOND PART OF THE SENTENCE SHOULD DESCRIBE THE DIFFERENCE BETWEEN THE SPECIES, AND NOT JUST A PROPERTY OF EHUX. "A SIGNIFICANT AMOUNT OF CaCO_3 " IS CONTAINED IN ANYTHING MADE OF CALCITE !!!!

The sentence

" For *E. huxleyi* P41, relation 3 does not stand any more as central area (i.e. the grid) of the coccoliths contains a significant amount of CaCO_3 ."

is restated :

"Amongst the seven species that we explored, *E. huxleyi* P41 is the only one for which the correlation between the number of segments and the mass of the coccolith is poor (see the circle in Fig.4E). This can be explained as the coccoliths *E. huxleyi* P41 have a such amount of CaCO_3 in the central area of that the mass of the grid is no more negligible compare to the masses in the tube and shield regions."

The very large amount of CaCO₃ in the central area of the coccoliths *E. huxleyi* P41 implies that the mass of the grid is no more negligible compared to the masses of the tube and shield regions.

187 Evolution... PLEASE USE ANOTHER TERM FOR THIS THROUGHOUT THE TEXT

Line 187, we remove "Evolution of the".

Lines 190, 192, 193 and 197, we remove "Evolution of the"

188 species, except *E. huxleyi* P41 for which p can't be determined. WHY ?

p is determined from the intersection between the grid and the tube. The grid is spread in the plane of the coccoliths whereas the out of plane thickness of the grid is generally very small. On the contrary, the out of plane thickness of the tube surrounding the grid is quite important. So the limit between grid and tube is generally easy to detect. For *E. huxleyi* P41, this limit is no more visible because the out of plane thickness of the grid is comparable to the one of the tube. In other words, it is not possible to determine the frontier between grid and tube and as a consequence the peripheral grid perimeter.

In order to clarify this point in the manuscript, we complete the sentence line 188 like this :

"... except *E. huxleyi* P41 for which p can not be determined because the out-of-plane thickness of the grid is comparable to the one of the tube rendering impossible the identification of the frontier between the 2 elements."

188 species, except *E. huxleyi* P41 for which p can't be determined. The yellow dotted line is $m = k_p \times p^\beta$ with m in pg, p in μm , $k_p = 4.92 \times 10^{-2}$, $\beta = 3.175$ (coefficient of determination $R^2 = 95\%$). PLEASE REPORT STANDARD DEVIATIONS. IS THE DEVIATION FROM 3 OF THE POWER PARAMETER SIGNIFICANT ?

This point was already explained and modified above.

190, 192, 193, 197 ... Evolution ... PLEASE USE ANOTHER TERM

Lines 190, 192, 193 and 197, we remove "Evolution of the"

191 given p , coccoliths with short rims display robust distal shields; coccoliths with long rims display delicate distal shields. PLEASE DEFINE THE TERMS "robust" AND "delicate" IN A QUANTITATIVE WAY.

The term "delicate" means that there is a slit (i.e. a space) between 2 neighborhood elements of the distal shields. On the contrary "robust" refers to elements of the distal shields in contact (no space).

To clarify this point in the manuscript, the following sentence :

"coccoliths with short rims display robust distal shields; coccoliths with long rims display delicate distal shields."

was replaced by

"coccoliths with short rims display robust distal shields (i.e. no space between the elements) while coccoliths with long rims display delicate distal shields (slit between the elements)."

196 tube at the periphery of the grid. For *G. muelleriae* and *E. huxleyi* P41, it was not possible to determine n and p , respectively. WHY ?

We replaced the following sentence

"For *G. muelleriae* and *E. huxleyi* P41, it was not possible to determine n and p , respectively."

by

"For *G. muelleriae*, the determination of n was not possible because the frontier between the successive segments is not resolved. For *E. huxleyi* P41, p cannot be measured (see text above)"

198 segment number n . The yellow dotted line is $m = k_n \times n^\beta$ with $k_n = 4.73 \times 10^{-5}$, m in pg. WHAT ABOUT BETA ? PLEASE GIVE STANDARD DEVIATIONS FOR ALL PARAMETERS.

As we said previously, line 165, the following sentence is added :

"It is worth noting that by assuming $\beta = 3$, the parameter $k_p = 6.67 \times 10^{-2}$ ($\pm 0.4 \times 10^{-2}$ with 95% confidence bounds) is obtained. "

Line 178, the sentence :

"with $k_n = 4.73 \times 10^{-5}$ and m in pg (fig. 4E)."

is replaced by

" with m in pg and $k_n = 4.73 \times 10^{-5}$ ($\pm 0.28 \times 10^{-5}$ with 95% confidence bounds) for $\beta = 3.175$ (fig. 4E). By fixing $\beta = 3.0$, the fit to the data with relation 3 yields $k_n = 9.39 \times 10^{-5}$ ($\pm 0.66 \times 10^{-5}$ with 95% confidence bounds). This shows that a small change in the exponent β has a big effect on k_n ."

Line 198, we replace :

" The yellow dotted line is $m = k_n \times n^\beta$ with $k_n = 4.73 \times 10^{-5}$, m in pg."

by

" The yellow dotted line is $m = k_n \times n^\beta$ with $k_n = 4.73 \times 10^{-5}$, $\beta = 3.175$ and m in pg."

200 ... contains significant amount of CaCO₃

As said previously, the sentence

"For *E. huxleyi P41*, relation 3 does not stand any more as central area (i.e. the grid) of the coccoliths contains a significant amount of CaCO₃."

is restated :

"Amongst the seven species that we explored, *E. huxleyi P41* is the only one for which the correlation between the number of segments and the mass of the coccolith is poor (see the circle in Fig.4E). This can be explained as the coccoliths *E. huxleyi P41* have a such amount of CaCO₃ in the central area of that the mass of the grid is no more negligible compare to the masses in the tube and shield regions."

The very large amount of CaCO₃ in the central area of the coccoliths *E. huxleyi P41* implies that the mass of the grid is no more negligible compared to the masses of the tube and shield regions.

201 ... (F) Scheme... THIS SCHEME ILLUSTRATES WHY THE SIZE OF THE INNER RING IS DETERMINED BY THE BASE PLATE. THIS IS NOT A NEW IDEA, AS YOU WILL ADMIT. BUT THIS SCHEME IN NO WAY ILLUSTRATES WHY THE FINAL (!) SIZE SHOULD CORRELATE WITH THE INITIAL SIZE.

We take into account this remark by adding on the scheme of Fig4.E that for mature coccoliths, the height of the tube is almost proportional to n (n is the number of segments), as well as the length of the rim. In such a way, it appears that m is almost proportional to n³.

In addition, lines 205-206, we replace the two sentences :

"In plane (forming the rim) and out-of-plane (forming the tube) growth of the coccolith. The bidimensional growth is proportional to the segment number n ."

by

"In plane (forming the rim of length L) and out-of-plane (forming the tube of height t) growth of the coccolith. With L and t almost linearly proportional to the segment number n and with w a constant equal to around 112 nm, the mass of each segment is almost proportional to n^2 . The symbol \sim refers to "almost linearly proportional to". This leads to a coccolith mass almost proportional to n^3 ."

203 unit nucleation sites, one every 112 nm in average I UNDERSTOOD YOU SAID THIS DEPENDS ON SPECIES ! ?

No, the average width of the segment is constant. To be clearer, we propose to add line 175 the following sentence :

Hence, the average width w of the segments appears to be a constant close to 112 nm whatever the species.

206 The bidimensional growth is proportional to the segment number n . PLEASE BE PRECISE ! HOW CAN "GROWTH" BE PROPORTIONAL TO AN INTEGER "NUMBER"

? IS IT GROWTH SPEED ? GROWTH RATE ? DOES THE RATE REALLY JUST JUMP IN STEPS OF INTEGER NUMBERS ?

Line 206, we replace

"

The bidimensional growth is proportional to the segment number n . "

by

" With L and t almost linearly proportional to the segment number n and with w a constant equal to around 112 nm, the average mass of each segment is almost proportional to n^2 . "

210 conditions, pH, salinity, carbonate chemistry to forecast the coccolithophorid biomineralization upon climatic changes PLEASE RESTATE ! WHAT WOULD BE THE MEANING OF "biomineralization upon changes" ?

Lines 209-2214, the sentence is restated :

Whereas previous researches focused on the dependence of the environmental conditions (i.e. pH, salinity, carbonate chemistry, ...) on the biomineralization of coccolithophorids, our work demonstrates that in normal condition of culture, the nucleation site number and the size of coccoliths depends strongly on the *obps* size. This in turn explains the important coccolith mass variability (with a coccolith mass ratio up to 3 within a single coccosphere) by the high *obps* size variability.

212 nucleation and the growth of a protococcolith and as a consequence the mass of a mature coccolith depends strongly on the *obps* size PLEASE RESTATE. NUCLEATION DOES NOT DEPEND ON THE *obps* SIZE. IT MAY BE THAT THE NUMBER OF NUCLEI DEPENDS ON THE *obps* SIZE. AGAIN, WHAT PRECISELY IS MEANT BY "GROWTH". GROWTH, AGAIN, DOES NOT DEPEND ON THE *obps* SIZE, I WOULD BET IT DEPENDS ON SUPERSATURATION AND MATERIAL TRANSPORT. IT MAY BE THAT THE FINAL MASS OF THE COCCOLITH DEPENDS ON THE ORIGINAL *obps* SIZE, BUT NOT "GROWTH". THE INACCURACY OF THESE STATEMENTS MAKES THE PAPER RATHER CONFUSING.

As we said in the previous answer, line 210, the sentence was restated. In particular, "*nucleation*" was replaced by "*nucleation site number*" and "*growth*" was replaced by "*size of coccoliths*".

214 within a coccosphere appears to be related to the *obps* size variability. THE "appears" MAKES THIS A RATHER VAGUE STATEMENT. IF THIS PORPOSITION IS CORRECT AT ALL, THE LOGICAL REASONING BEHIND IT NEEDS TO BE DESCRIBED.

Two answers before, we already modified this point. We added the sentence :

" This in turn explains the important coccolith mass variability (with a coccolith mass ratio up to 3 within a single coccosphere) by the high *obps* size variability."

230 our new *obps* size - coccolith mass framework established the size of *obps* (and therefore

coccolith mass) is presumably largely regulated by the cell size and the cell nucleus size, I DO NOT UNDERSTAND THIS SENTENCE. WHAT IS A "size-coccolith" ? WHAT IS A "mass framework" ? TO WHICH SUBJECT DOES THE PREDICATE "is" REFER ? ETC..

Thank you for raising this issue. We agree that the formulation requires better clarity. To understand this sentence, you have to associate the terms "obps size" and the terms "coccolith mass". We restate the sentence like this :

"our newly established correlation between *obps size and coccolith mass* suggests that the size of *obps* (and therefore coccolith mass) is presumably largely regulated by the cell size and the cell nucleus size,"

1 The dominant role of the organic base plate scale in the mass of coccoliths revealed by X-ray tomography. THIS TITLE SAYS LITERALLY THAT THE (MASS) OF THE BASE PLATE SCALE DOMINATES THE MASS OF THE COCCOLITHS. HOWEVER, THE AUTHORS DO NOT APPEAR TO MEAN THIS. OR ? "DOMINATION" IS NOT A VERY APPROPRIATE TERM HERE ANYWAY, UNLESS OTHER FACTORS DETERMINING THE MASS OF THE COCCOLITHS WERE MEASURED AND DISCUSSED. ONLY THEN THE AUTHORS WOULD BE ABLE TO SHOW THAT THE SIZE(!?) OF THE BASE PLATE SCALE DOMINATES THE OTHER FACTORS.

We understand that the term "dominant" is probably not appropriate as it may be interpreted as the dominance of the obps compared to other factors

Line 182, the sentence

" Fig. 4. The dominant role of the *obps* in the nucleation and growth of the coccoliths."

was replaced by

" Fig. 4. Dependence of the *obps* size on the number of segments and the mass of coccoliths."

In addition, even if there are some evidences that the size of the grid and the size of the obps are correlated, we prefer in the title to insist on the morphological features directly observable by tomography : grid size, segment number and mass. We propose to restate the title as :

"Positive correlation between grid size, segment number and mass of coccoliths revealed by X-ray tomography of single coccolithophores"

19 with a resolution close to 30 nm. After isolating... PLEASE EXPRESS THIS MORE PRECISELY. DID YOU ISOLATE COCCOLITHS IN THE IMAGES OR DID YOU ISOLATE THEM BY "MECHANICAL" PREPARATION

We replace :

"After isolating"

by

“After segmenting each coccolith from the coccospheres, “

21 perimeter. Assuming that this area is reminiscent of the organic base plate scale...
PLEASE SUBSTANTIATE THIS ASSUMPTION AND REPLACE “reminiscent” BY A
MORE APPROPRIATE WORD

The coccolithogenesis begins with the mineralisation of calcite along the rim of the obps. This has been known for a long time (Mann et al, 1988). Thus, the equality between the perimeter of the central area and the perimeter of the obps is not an assumption. We restate by replacing this sentence :

" Assuming that this area is reminiscent of the organic base plate scale *obps* on which the biomineralization was initiated, our findings reveal the dominant role of the *obps* in the CaCO₃ nucleation site number and the growth of coccoliths, strengthening the interplay between photosynthesis and biomineralization. "

by

" As this area is “expected to be” the mark of the organic base plate scale *obps* around which the biomineralization was initiated, our findings highlight the role of the *obps* size on the number of CaCO₃ nucleation sites and the mass of coccoliths, and strengthen the interplay between photosynthesis and biomineralization."

23 CaCO₃ nucleation site number and the growth of coccoliths... PERHAPS “number of nucleation sites” WOULD BE EASIER TO UNDERSTAND

We agree. We replace "CaCO₃ nucleation site number" by " number of CaCO₃ nucleation sites"

23 CaCO₃ nucleation site number and the growth of coccoliths, strengthening the interplay between photosynthesis and biomineralization. PLEASE RESTATE: WHO STRENGTHENS WHAT ? THE COCCOLITHS CERTAINLY DO NOT STRENGTHEN THIS INTERPLAY.

This is "our findings" which strengthen the interplay between photosynthesis and biomineralization.

We replace "strengthening" by "and strengthen"

Reviewer #2 (Remarks to the Author):

I read with a lot of attention the manuscript presented by Beuvier et al., entitled “The dominant role of the organic base plate scale in the mass of coccoliths revealed by X-ray tomography”.

This work presents a careful analysis of the relationships between the sizes of several specific structural features of a coccolith and between the coccolith structural features and its organic base plate scale. It evidences the role of the cell and cell nucleus sizes in the biomineralization process of these marine species. The conclusions are robust, based on an extended set of data, including several coccolith species and corresponding to a considerable amount of experimental results. The supplementary data are very helpful in order to follow all the steps

of the analysis process, obviously performed in a very careful and systematic way. The whole article is very well written and should be readable by a large audience.

In addition, it makes no doubt that these original findings could only be obtained by the use of 3D x-ray CDI, which provides in a non destructive way, a 3D representation of the individual coccolith volumes, while minimizing the manipulation of these fragile biominerals. Neither optical nor electron microscopy nor x-ray scanning diffraction could provide equivalent results.

For all these reasons, I think that this manuscript is suitable for Nature Communications, provided that the authors consider the few questions, reported below, in order to improve the strength of the manuscript.

1 – Resolution estimation: the resolution estimation is presented in the Supplementary material for *E. Huxleyi*, one of the largest of the analyzed coccoliths. First, I think that the shown PRTF is integrated over all directions, while I think that the effective resolution in one direction should rather be presented. Second, I would like to see the resolution estimation for all species, including the smaller one, where one may expect less scattered photons (unless the acquisition time has been scaled accordingly).

In CXDI the obtained resolution is determined by the largest scattering vector where the intensity is observed. The number of scattered photons depends on optical contrast (the same for all samples), the size of the specimen (scattering volume) and exposure time. We used between 4.5 and 25 seconds of exposure time. The studied specimens turn out to be strong scatterers. Independent on the sample size or exposure time used the measured 2D diffraction patterns were covered by speckles to the edge of the detector. As a result the resolution in this study was mainly limited by the detector size. This point is now explained in supporting information. In addition, the figure S2 which shows PRTF for one species was replaced by the following figure

This new figure shows the PRTFs for 3 species. The voxel size is $28.8 \times 28.8 \times 28.8 \text{ nm}^3$ for *R. parvula* and *E. huxleyi* RCC1216. At a resolution of 28.8 nm, the PRTFs are higher than 0.5. The real resolutions are thus close to the voxel size for the experiment done on these 2 species. For *G. oceanica*, the voxel size is $32.5 \times 32.5 \times 32.5 \text{ nm}^3$. For PRTF = 0.5, the real resolution is equal to 35.0 nm. This value is thus slightly higher than the voxel size. The text in supporting information was modified accordingly.

In addition, the reviewer would like to see the resolution estimation in one direction. We think that such estimation could be biased depending on the direction chosen and the sample edge at which the resolution is estimated. Therefore, we believe that the PRTF is a better estimate of the resolution as it takes into account all measured directions and sample features.

2 – line 68, the typical size of the coccoliths should be given, as the argument directly refers to the resolution capability of 3DCDI.

Line 68, we replace :

" hence the technique is well suited to determine the mass of individual coccoliths. "

by

" hence the technique is well suited to image coccospheres of 1 - 7 micrometer size and to determine the mass of individual coccoliths. "

3 – Please unify the terminology associated to the method: it is sometimes referred to as 3DCDI or to x-ray tomography. I think 3D CDI is more appropriate.

X-ray tomography is a general term. It could refer to several techniques :

- X-ray tomography by absorption contrast,
- X-ray tomography by phase contrast
- ptychography
- 3D-CXDI
- ...

So 3D-CXDI is one of the technique of X-ray tomography.

To clarify this point, line 17, the sentence :

" Synchrotron three-dimensional Coherent X-ray Diffraction Imaging was used to image coccolithophores of the *Gephyrocapsa*, *Emiliana* and *Reticulofenestra* families with a resolution close to 30 nm. "

is replaced by

" Synchrotron three-dimensional Coherent X-ray Diffraction Imaging, one of the technique of nanotomography, was used to image coccolithophores of the *Gephyrocapsa*, *Emiliana* and *Reticulofenestra* genera with a resolution close to 30 nm. "

In addition, "families" is now replaced by "genera".

4 – Supplementary material: line 83 “blue curve” should be “purple curve”

It is done.

Reviewer #3 (Remarks to the Author):

The manuscript by Beuvier et al employs a novel approach to determining the mass of coccoliths of three ecologically and evolutionarily important coccolithophore species. The novel application of X-ray tomography allows accurate determination of the volume of individual coccoliths along with detailed 3-D structure, allowing the relative mass proportions of different regions of the coccolith to be measured. The authors have provided a convincing validity analysis of the approach and the data that is presented is also convincing.

The main conclusions are: 1). The size and curvature of the nucleus determines the size and curvature of the coccoliths, and it is proposed that variations in the size of the nucleus with cell cycle account for the observed coccolith size distributions. This largely confirms what previous studies have also proposed. 2). The mass of the coccolith is determined by the number of coccolith segments in the central tube area perimeter. It is concluded that this in turn is determined by the number of crystal nucleation sites provided by an organic base plate scale.

While the data are convincing, I think that the manuscript could have been strengthened by the provision of addition data to support the conclusions. Direct determination of nuclear size would have allowed a more robust relationship between nucleus and coccolith to be made. It should not be difficult to obtain such measurements.

To observe the ultrastructure and in particular the cell nucleus with a good resolution, the use of transmission electron microscopy on chemically fixed cells is the most common technique (van der Wal 1983, Westbroek 1984, Yin et al 2018). Images are obtained on sections. Hence, this technique is a destructive technique. By using cryo-FIB SEM, Sviben et al (Nat. Com. 2015) were able to image the cell nucleus in 3 dimensions. However, this technique is not simple to set up as the sample has to be vitrified before being observed slice by slice. This technique renders possible the observation of both the coccoliths and the cell nucleus. As TEM, it is also a destructive technique.

For these 2 techniques, the extracted information is the size of the cell nucleus, the size of the protococcolith and the size of the mature coccoliths of the coccosphere. A correlation can be done between the size of the protococcolith and the size of the cell nucleus, But, it is not possible to correlate the diameter of the cell nucleus with the size of the coccoliths of the coccosphere.

To make a direct correlation between the cell nuclear size and the coccolith size, cells have to be observed *in-situ* during the coccolithogenesis (the cell should be alive) : when a cell produces a coccolith, the measurement of the cell nucleus should be done. This is technically not feasible, neither by TEM, nor FIB-SEM, nor X-ray tomography.

The conclusion that the obps structure underlies this structure is postulated in a number of other publications. However, very little is known about the nucleating so-called baseplate structures in coccolithophores, particularly the species under study. The conclusions are, therefore somewhat speculative and I am not sure if the conclusion implicit in the title “The dominant role of the organic base plate scale in the mass of coccoliths revealed by X-ray tomography” is fully supported by the data provided since no information on the actual obps is provided.

We agree that the *obps* was not visible during our experiment and that indirect measurements of the *obps* size were deduced from the grid size of the coccoliths. However, recent articles confirm the presence of the organic base plate. See for instance the article of Yin 2018 :

FIG. 4. TEM micrograph of a chemically fixed *Emiliaia huxleyi* cell showing a coccolith vesicle (cv) with remains of calcite (arrow). Note that most mineral is dissolved during the procedures for fixation and embedding; cw, cell wall; cl, chloroplast; G: Golgi apparatus; Gv, Golgi vesicles; M, mitochondrion; N, nucleus; py, pyrenoid; rb, reticular body.

The base plate is visible (black arrow) :

This reference was added line 81.

To take into account the remark of the referee, we propose to restate the title like this:

“Positive correlation between grid size, segment number and mass of coccolith revealed by X-ray tomography of single coccolithophores”

Reviewers' comments:

Reviewer #1 (Remarks to the Author):

Review of NCOMMS-18-02012A

The high scientific value of the paper was described in my previous review so that I will not repeat this part of a review here.

The manuscript has been improved considerably and I have only some minor comments left.

The minor comments are mainly related to language and terminology, which should be easy to amend or clarify.

Abstract

"one of the technique of nanotomography" "technique" should be "techniques"

"our findings highlight the role of the obps size on the number of CaCO₃ nucleation sites and the mass of coccoliths, and strengthen the interplay between photosynthesis and biomineralization."

Most of the abstract state issues which are well-known. The actual contents of the paper is only addressed with this non-committal qualitative statement. This is not suitable to attract attention as it hardly reflects the contents of the paper. The authors should state their results quantitatively.

The results do not strengthen the interplay between photosynthesis and biomineralization. The results may strengthen the view that there is an interplay between photosynthesis and biomineralization.

But I think that this most non-committal statement is also absolutely trivial. If there is no photosynthesis the organism will not biomineralize. So it is quite clear that there is an interplay. In the abstract of a paper you need to state WHAT IS THE INTERPLAY.

Main Text

The authors should be fine their angle alpha in a drawing. The cross sections in Fig.2, to which the text refers when introducing the angle alpha, appear to be quite circular arcs, such that it does not become clear how this angle is measured. Also, the other parameters like p, t etc. L should be unequivocally defined in words and as abbreviation in a sketch in the supplementary material. At present, it is left to guess for the reader what is exactly what in the terminology. When reading the previous version of the paper I believed that the "perimeter" corresponds to the red line in the insert in Fig. 4D, now I see that it may be the inner perimeter which is indicated in green in Fig. 4D, i.e. the perimeter of the organic base plate? The confusion arises from the fact that the term "grid" is not defined in the paper. I am sure this can all be made clear in one simple sketch.

Line 84 quotation mark " missing

lines 107-108: Sentence unclear. What is meant by "the one of the obps". Do you mean "these eccentricities originate from the obps disc on which the nucleation of the coccolith took place" ?

Line 111: reference 40 should also be referenced at the end of this line.

Line 144 "variability between a population" . As the article "a" is a singular case it is unclear what is the variability referring to. Do you mean "variability within a population" ?

Line 153 "obeying to the formula" should be "obeying the formula"

Line 157: Logic? "This relation means that the mass of a coccolith m is to a first approximation determined by the external perimeter of the grid p." Stated like this this means essentially $m = p$. But this is clearly not meant. There are also the parameters k_p and β . I know they are fit parameters. But what determines their values as physical parameters. What are their physical meaning? How do they correlate with the parameters L, t, and n which are mentioned in Fig. 4. Why do "robust" and "delicate" shields follow the same equations (1) and (3) although they are so different in Fig. 4B?

Line 184: Plural/Singular mixed in "the R-units" ?

Line 185-188 Please restate. I do not understand the term "vestigial" in a context, which suddenly divides w in half. W is one of the few parameters which appeared to be clearly defined ($w = p/n$?). So why is there suddenly an uncertainty of 50%. Please explain what you mean here in more detail. Most importantly, the observation that w is "a constant close to 112 nm whatever the species" is perhaps the most significant observation with respect to the nucleation of coccoliths, so it should not be in a confusing context that it is also 55-60 when considering something "vestigial".

Line 193 "this shows that a small change in the exponent beta has a big effect on kn ." Please restate: "this shows that the exponent beta and the prefactor kn are highly correlated in the fit."

Line 194 Mathematics ? Terminology ? "As the exponent beta > 2 , relation (3) means that the nucleation site number n and the average mass of each segment m/n are positively correlated." Well, let us suppose beta = 2 or $m = kn * n^2$, then $m/n = kn * n$, and as kn is positive, they would also be positively correlated. So I do not understand the meaning of this sentence. Perhaps "positive correlation" has different meanings in statistics and physics ?

Line 201 "compare" should be "compared" (same mistake also in lines 91 and 109 of supplement)

Line 201-202 In this case the mass of the grid is no more negligible compare(d) to the masses in the tube and shield regions. Please explain this in more detail. Do you neglect the mass in the grid in the calculations for the other species ? Why ? Does this put a big question mark on the whole procedure ?

Line 222 Add an "a": "contains a significant amount"

Line 230 Terminology ? "almost linearly proportional". I believe in Mathematics "proportional" does mean that there is a linear relationship. So the word "linearly" can be dropped. If the relationship is non linear, it is not proportional, but e.g. a monotonous function. If, e.g. $y(x) = a x^3$, y is proportional to x^3 . Please check !

Line 232-236 While the method introduced by the paper appears to be brilliant, and all the conclusions are correct and sound, I am still unhappy with the logic of the discussion. The half-sentence "whereas previous studies focused on ..." is logically untrue (as long as which previous studies is not specified) and, moreover, irrelevant in the discussion. Simply delete this. The half sentence: "the nucleation site number and the size of coccoliths depends strongly on the obps size". First, please define "nucleation site number". Second, since the obps has not been observed in this study, the discussion should not make such plain statements about the obps. The discussion should be anchored at the new observations made in the present study and discuss them in the light of the literature. And please make clear what is observed and what is concluded. You may consider to start with something like: Our study clearly showed that number of coccolith segments scales linearly with the perimeter length of the XXX[please insert a in this paper well defined term here], and the segments occur at a constant distance of $112 \pm x$ nm, whatever the species. According to [REFERNCELiterature] nucleation of the segments occurs on the outer perimeter (?) of the organic base plate scale within the coccolith vesicle. We thus conclude that the nucleation sites for the segments on the organic base plate scale are at a constant distance of $112 \pm x$ nm on the outer perimeter of the organic base plate scale, independent of the actual size of the obps. An increasing or decreasing size or perimeter length of the obps is accommodated by the production of more or fewer, respectively, segments.

Line 236: "coccolith mass ratio" A ratio implies two quantities. Exactly which two quantities do you mean ?

Line 237/238: An angle is measured between two vectors. How is this angle measured ? Which are these vectors ?

Line 239: Please make a full stop after coccolith size.

Line 252: "established". I do not agree that this is established. It is proposed as indirect conclusion.

Line 253: "correlation ... mass framework...". Delete "framework". It makes no sense in this sentence. ?

Line 255: The final sentence makes a very weak statement. Please consider to delete it.

Hoffmann et al. (2014) must be referenced.

Hoffmann et al. discuss the measurement of coccosphere mass and not the coccolith mass. So, indeed, it does not provide new method to determine the coccolith mass. You (the authors) appear to argue that you restrict your considerations to coccolith mass, yet you reference a lot of papers in the introduction which are even not related to coccolithophores at all. So why not reference a significant paper on coccosphere mass ? I do not accept your argument.

Reviewer #2 (Remarks to the Author):

I am reviewing the manuscript for a second time and I am fully satisfied with the modifications and additional information brought by the authors.

I recommend publication in Nature Communications.

Reviewer #3 (Remarks to the Author):

The manuscript by Beuvier has been substantially revised and improved.

In relation to my comments, I accept that obtaining direct detailed comparison of nuclear and coccolith size is difficult. The authors correctly argue that providing this information with EM techniques is not possible since the newly formed coccoliths need to be correlated with the size of the nucleus at the time of production. However, it should be possible to provide information that supports the conclusions about nuclear size by simply measuring the size of the nucleus during the cell cycle using a fluorescent vital dye - of which there are a number of options available. Confocal sectioning should be able to provide data on the changes in nuclear size even with the smaller *E. huxleyi* cells and certainly with the larger species. Parallel monitoring of the size of newly produced coccoliths by using negative staining with calcein (staining the whole coccosphere, washing and observing the unstained new coccoliths) should be able to provide a time course of coccolith size during the cell cycle.

Reviewers' comments:

Reviewer #1 (Remarks to the Author):

Review of NCOMMS-18-02012A

The high scientific value of the paper was described in my previous review so that I will not repeat this part of a review here.

The manuscript has been improved considerably and I have only some minor comments left. The minor comments are mainly related to language and terminology, which should be easy to amend or clarify.

Abstract

“one of the technique of nanotomography” “technique” should be “techniques”

Now this is correct.

“our findings highlight the role of the obps size on the number of CaCO₃ nucleation sites and the mass of coccoliths, and strengthen the interplay between photosynthesis and biomineralization.”

Most of the abstract state issues which are well-known. The actual contents of the paper is only addressed with this non-committal qualitative statement. This is not suitable to attract attention as it hardly reflects the contents of the paper. The authors should state their results quantitatively. The results do not strengthen the interplay between photosynthesis and biomineralization. The results may strengthen the view that there is an interplay between photosynthesis and biomineralization. But I think that this most non-committal statement is also absolutely trivial. If there is no photosynthesis the organism will not biomineralize. So it is quite clear that there is an interplay. In the abstract of a paper you need to state WHAT IS THE INTERPLAY.

The main idea in the relationship between photosynthesis and biomineralization is that the photosynthesis drives the formation of the *obps* (the larger the size of the obps, the greater the photosynthetic activity) whereas biomineralization is related to the mass of the coccolith. To avoid confusion, the part of the sentence "and strengthen the interplay between photosynthesis and biomineralization" is removed.

We have significantly modified the summary. We now find it more striking by respecting the 150-word maximum rule.

Main Text

The authors should be fine their angle alpha in a drawing. The cross sections in Fig.2, to which the text refers when introducing the angle alpha, appear to be quite circular arcs, such that it does not become clear how this angle is measured.

A scheme of the alpha angle (α) is specified in Figure 2C and in figure S11.

Line 109, “both shields of the coccoliths are out-of-plane, inclined by about $\alpha \sim 30 \pm 5^\circ$ ” is replaced by “both shields of the coccoliths are out-of-plane inclined by about $\alpha \sim 30 \pm 5^\circ$ ”. In others words, the coma is deleted.

Also, the other parameters like p , t etc. L should be unequivocally defined in words and as abbreviation in a sketch in the supplementary material. At present, it is left to guess for the reader what is exactly what in the terminology. When reading the previous version of the paper I believed that the “perimeter” corresponds to the red line in the insert in Fig. 4D, now I see that it may be the inner perimeter which is indicated in green in Fig. 4D, i.e. the perimeter of the organic base plate? The confusion arises from the fact that the term “grid” is not defined in the paper. I am sure this can all be made clear in one simple sketch.

Line 86, p is defined as a function of a and b , the semi-major axis and the semi-minor axis of the grid. For sake of clarity, we replace a and b by a_g and b_g , the major axis and the minor axis of the grid. These definitions are now in agreement with table 1 in supplementary information and in agreement with the new figure S11.

Line 159, L is defined as “the length of the proximal rim”.

Line 167, t is defined as the “thickness of the tube”.

Line 172, n is defined as the “number of calcite segments”.

Line 175, w is defined as the tube average tangential width of the calcite segments at the periphery of the grid.

So, there are 6 important parameters : α , p , L , t , n and w .

Line 169, we add the following sentence :

“For the sake of clarity, the parameters a , b , a_g , b_g , α , p , L , t and w are schematized in Supplementary Figure 11.”

Supplementary Figure 11 corresponds to the following figure :

Line 84 quotation mark “ missing

The quotation mark “ is now added.

lines 107-108: Sentence unclear. What is meant by “the one of the obps”. Do you mean “these eccentricities originate from the obps disc on which the nucleation of the coccolith took place “ ?

Yes. To take into account this remark, we replace the following sentence :

“These eccentricities may originate from the one of the *obps* on/around which the nucleation of the coccolith took place.”

by this sentence :

“These eccentricities may originate from the elliptical shape of the *obps* on/around which the nucleation of the coccolith took place during the formation of the protococcolith.

Line 111: reference 40 should also be referenced at the end of this line.

We agree with the reviewer and this is now done. Reference 40 refers to the work of Yin et al. where it is written that

“[] as it apposed to the nuclear envelope, the organic template develops on a curved surface. Hereby the crystallographic alignment of each R-unit is influenced by the curvature of the nuclear envelope”

Line 144 “variability between a population” . As the article “a” is a singular case it is unclear what is the variability referring to. Do you mean “variability within a population” ?

Line 144, the sentence :

“This shows that a large part of this variability comes from variability between a population of coccoliths coming from individual coccospheres.”

is replaced by this one :

“This shows that the distribution of coccolith masses within a coccosphere is indicative of the distribution of masses within a species.”

Line 153 “obeying to the formula” should be “obeying the formula”

Now, “obeying to the formula” is replaced by “obeying the formula”

Line 157: Logic ? “This relation means that the mass of a coccolith m is to a first approximation determined by the external perimeter of the grid p .” Stated like this this means essentially $m = p$. But this is clearly not meant. There are also the parameters k_p and β . I know they are fit parameters. But what determines their values as a physical parameters. What are their physical meaning ? How do they correlate with the parameters L , t , and n which are mentioned in Fig. 4. Why do “robust” and “delicate” shields follow the same equations (1) and (3) although they are so different in Fig. 4B ?

Lines 157-158, to take into account the first remark of the reviewer, the following sentence :

“This relation means that the mass of a coccolith m is to a first approximation determined by the external perimeter of the grid p .”

is replaced by

“Indeed, the relation (1) means that the mass of a coccolith m is directly linked to the perimeter p of the grid, i.e. when p is known the mass of the coccolith can be estimated from (1). As observed in Fig. 2D, this can be understood because an important part of the mass of a coccolith is located in the tube region.”

We replace also replace the following sentence (lines 158-160) :

“In addition, there is an interplay between shield rim robustness and length (Fig. 4B). The length of the proximal rim L scales linearly with p within species, but with a different constant of proportionality for different species.”

By

“By looking in more detail, we observe also that the length of the proximal rim scales linearly with p within species, but with a different constant of proportionality for different species.”

Line 156, due to the high uncertainty of k_p , we replace “ $k_p = 0.04925 \pm 0.02170$ ” by “ $k_p = 4.92 \pm 2.17$ ”.

Now, what about the meaning of k_p and β ? As explained lines 159-160, L is proportional to p . And as explained line 167, t is proportional to p . So, these observations may explain why β is close to 3. This is detailed lines 167-168-169. To understand the meaning of k_p , let's have a comparison with the body mass index (BMI). The BMI is defined as the body mass divided by the square of the body height. The BMI gives a value which allows to know whether a person is underweight, normal weight or overweight. So, by analogy, we can considered that k_p is the coccolith mass index. However, k_p is quite constant whatever the species. In other words, we were unable to identify different k_p values for the different species, probably because the mass is mainly located in the tube, which is a common feature for all the species. To identify differences between varieties, we looked at rim lengths L and tube heights t , and it was by measuring these parameters that differences were identified.

To help reader to understand the meaning of k_p , line 157, we add the following sentences :

“ k_p may be called the coccolith mass index. Amazingly, we can consider that all the species have the same index.

We understand that the reviewer is surprised by the fact that for a given p , the mass of the coccoliths with robust shields is similar to the ones having delicate shields. This is already mentioned and explained lines 162-164 :

“Thus, CaCO_3 biomineralization takes place by favoring the growth of either the distal part with calcification between segments (case of *G. oceanica*) or the proximal part with longer proximal rims (case of *E. huxleyi* RCC1212).”

Line 184: Plural/Singular mixed in “the R-units” ?

“due to the overgrowth of the R-units crystals”

is replaced by

“due to the overgrowth only of the R-unit crystals”.

Line 185-188 Please restate. I do not understand the term “vestigial” in a context, which suddenly divides w in half. W is one of the few parameters which appeared to be clearly defined ($w = p/n$?). So why is there suddenly an uncertainty of 50%. Please explain what you mean here in more detail. Most importantly, the observation that w is “a constant close to 112 nm whatever the species” is perhaps the most significant observation with respect to the nucleation of coccoliths, so it should not be in a confusing context that it is also 55-60 nm when considering something “vestigial”.

As we explained from the line 179, each segment of the coccoliths is composed of 2 types of calcite crystals, one with the c-axis orientation parallel to the coccolith plane and denoted R-unit (“R” for radial) and the other with the c-axis perpendicular to the coccolith plane (V-unit; “V” for vertical)^{44,48}. Even if the initial crystals of the proto-coccoliths are deposited with alternating radial and vertical c-axis orientation (the V/R nucleation model)^{46,49}, the volume of mature coccoliths is mainly composed of R-unit crystals in *Emiliana* and *Gephyrocapsa* genera due to the overgrowth only of the R-unit crystals. So the main idea is that the periphery of the *obps* is composed of R-unit nucleation sites and V-unit nucleation sites. The distance between 2 successive R-unit nucleation sites is equal to w . And the distance between 2 successive V-unit nucleation sites is also equal to w . So as a consequence, the distance between 2 successive CaCO_3 nucleation sites is equal to $w/2$. To avoid any confusion, line 187, we remove the sentence :

“(or every 55-60 nm if the vestigial V-unit nucleation sites are included).”

For sake of clarity, we replace the sentences of the lines 181-188 by the following ones :

“During the coccolithogenesis, the periphery of the *obps* is composed of alternating V-units and R-units^{42,50}. However, in mature coccoliths of *Reticulofenestra*, *Gephyrocapsa* and *Emiliana* genera, the volume is mainly composed of R-units because V-units are not developed⁵⁰. Thus, our findings let us to propose that the periphery of the *obps* controls the

mineralization site number n , with a R-unit nucleation site every w and also a V-unit nucleation site every w . As V-units are not developed, the average width w of the R-unit segments appears to be a constant close to 112nm whatever the species.“

Line 193 “this shows that a small change in the exponent beta has a big effect on kn .” Please restate: “this shows that the exponent beta and the prefactor kn are highly correlated in the fit.”

It is done.

Line 194 Mathematics ? Terminology ? “As the exponent $\beta > 2$, relation (3) means that the nucleation site number n and the average mass of each segment m/n are positively correlated.” Well, let us suppose $\beta = 2$ or $m=kn * n^2$, then $m/n = kn * n$, and as kn is positive, they would also be positively correlated. So I do not understand the meaning of this sentence. Perhaps “positive correlation” has different meanings in statistics and physics ?

Thank you for pointing out this error. We replace :

“As the exponent $\beta > 2$ ”

by

“As the exponent $\beta > 1$ ”.

Line 201 “compare” should be “compared” (same mistake also in lines 91 and 109 of supplement)

These 3 errors have been corrected.

Line 201-202 In this case the mass of the grid is no more negligible compare(d) to the masses in the tube and shield regions. Please explain this in more detail. Do you neglect the mass in the grid in the calculations for the other species ? Why ? Does this put a big question mark on the whole procedure ?

Of course, we never neglect or subtract the mass of the grid in the calculations. But, the mass of the grid is generally small compared to the mass of the remaining part of a coccolith except for one specie (*E. huxleyi P41*). So the procedure is valid except in the case of *E. huxleyi P41*. We think that our explanations are quite clear. So we just replace

“the mass of the grid is no more negligible compared to the masses in the tube and shield regions”

by

“the mass of the grid is no more small compared to the masses in the tube and shield regions”

The term “small” may be less misleading.

Line 222 Add an “a”: “contains a significant amount”

This is done.

Line 230 Terminology ? “almost linearly proportional”. I believe in Mathematics “proportional” does mean that there is a linear relationship. So the word “linearly can be dropped. If the relationship is non linear, it is not proportional, but e.g. a monotonous function. If, e.g. $y(x) = a x^3$, y is proportional to x^3 . Please check !

This is true. We remove “linearly”.

Line 232-236 While the method introduced by the paper appears to be brilliant, and all the conclusions are correct and sound, I am still unhappy with the logic of the discussion. The half-sentence “whereas previous studies focused on ...” is logically untrue (as long as which previous studies is not specified) and, moreover, irrelevant in the discussion. Simply delete this.

We remove this half-sentence.

The half sentence: “the nucleation site number and the size of coccoliths depends strongly on the obps size”. First, please define “nucleation site number”. Second, since the obps has not been observed in this study, the discussion should not make such plain statements about the obps. The discussion should be anchored at the new observations made in the present study and discuss them in the light of the literature. And please make clear what is observed and what is concluded.

You may consider to start with something like: Our study clearly showed that number of coccolith segments scales linearly with the perimeter length of the XXX[please insert a in this paper well defined term here], and the segments occur at a constant distance of $112 \pm x$ nm, whatever the species. According to [REFERNCELiterature] nucleation of the segments occurs on the outer perimeter (?) of the organic base plate scale within the coccolith vesicle. We thus conclude that the nucleation sites for the segments on the organic base plate scale are at a constant distance of $112 \pm x$ nm on the outer perimeter of the organic base plate scale, independent of the actual size of the obps. An increasing or decreasing size or perimeter length of the obps is accommodated by the production of more or fewer, respectively, segments.

We agree with this point of view. We replace the two first sentences of the Discussion section by

“Our study clearly showed that the number of R-unit segments scales linearly with the perimeter length p of the grid (see the sketch in Supplementary Fig. 11 for more details), leading to an average width of the segments of 110-120 nm, whatever the species. According

to literature^{38,40,43,52}, nucleation of the CaCO₃ segments occurs on the outer perimeter of the organic base plate scale *obps* within the coccolith vesicle. We thus conclude that the CaCO₃ nucleation sites are at a constant distance of 110-120 nm on the outer perimeter of the organic base plate scale, independent of the actual size of the *obps*. An increasing or decreasing size or perimeter length of the *obps* is accommodated by the production of more or fewer segments, respectively.

Below we put sentences coming literature highlighting that the nucleation of calcite appears at the periphery of the organic base plate.

“The plate was already present prior to calcification (see Fig. 9 in Klaveness 1972a) and probably provided the sites for incipient crystal growth on its rim” [van der Wal et al 1983]

“Nucleation produces a protococcolith ring of calcite (CaCO₃) crystals around the margin of a precursor organic base-plate” [Young 1992]

“Rhombohedra crystals are formed with an organic ‘base plate’ subtended between them” [Westbroek 1984]

“In *E. huxleyi*, the first crystals to appear are arranged along the periphery of the base-plate, where they form a wreath-shaped structure, the protococcolith ring, which is composed of single calcite crystals” [Paasche2011]

Line 236: “coccolith mass ratio” A ratio implies two quantities. Exactly which two quantities do you mean ?

Line 236, we replace :

“This in turn explains the important coccolith mass variability (with a coccolith mass ratio up to 3 within a single coccosphere) by the high *obps* size variability.”

by

“This in turn explains the important coccolith mass variability (with a mass ratio up to 3 between the lighter and the heavier coccoliths within a single coccosphere) by the high *obps* size variability.”

Line 237/238: An angle is measured between two vectors. How is this angle measured ? Which are these vectors ?

The meaning of the angle is now explained in figure 2 and supplementary figure 11.

Line 239: Please make a full stop after coccolith size.

We replace this sentence :

“As the out-of-plane inclination of the shields of the coccoliths is constant (inclined by about $\alpha\sim 30\pm 5^\circ$ along the major axis and $\alpha\sim 25\pm 5^\circ$ along the minor axis (Fig. 2B and C)) whatever the coccolith size, the *obps* size may be determined by the cell nucleus size, which varies significantly through the cell growth/division cycle.”

by 2 sentences :

“Our work shows also that the out-of-plane inclination of the shields of the coccoliths is constant (inclined by about $\alpha\sim 30\pm 5^\circ$ along the major axis and $\alpha\sim 25\pm 5^\circ$ along the minor axis (Fig. 2B and C) compared to the coccolith plane) whatever the coccolith size. We therefore speculate that the *obps* size could be determined by the cell nucleus size, which varies significantly through the cell growth/division cycle.”

Line 252: “established”. I do not agree that this is established. It is proposed as indirect conclusion.

The sentences of the lines 251 to 256 were replaced by :

“Even though the positive correlation between the cell size and the coccolith size was already reported within and between species⁴³⁻⁴⁴, further analysis using coherent X-ray diffraction imaging at cryogenic temperature on frozen-hydrated cell⁴⁵⁻⁴⁶ or confocal microscopy on stained cells would be needed to check whether the cell nucleus size, which varies significantly through the cell growth/division cycle, could regulate the size of *obps* and therefore the coccolith mass.”

Line 253: “correlation ... mass framework...”. Delete “framework”. It makes no sense in this sentence. ?

The sentence was modified (see above).

Line 255: The final sentence makes a very weak statement. Please consider to delete it.

Yes. It is done (see above).

Hoffmann et al. (2014) must be referenced.

Hoffmann et al. discuss the measurement of coccosphere mass and not the coccolith mass. So, indeed, it does not provide new method to determine the coccolith mass. You (the authors) appear to argue that you restrict your considerations to coccolith mass, yet you reference a lot of papers in the introduction which are even not related to coccolithophores at all. So why not reference a significant paper on coccosphere mass ? I do not accept your argument.

As proposed by the reviewer, line 55, we add the reference of Hoffmann, *Insight into Emiliana huxleyi coccospheres by focused ion beam sectioning*, Biogeosciences, 12, 825-834, 2015.

(1) Note that it is difficult to cite all the articles related to the measurement of coccosphere mass and coccolith mass.

(2) Note also that the reference of Hoffmann2014 is already included line 48 to speak about the crystallographic orientation of calcite in coccoliths.

(3) All the references related the paragraph dedicated to the measurement of coccosphere mass and coccolith mass (from lines 55 to 62) speak about coccolithophores.

Reviewer #2 (Remarks to the Author):

I am reviewing the manuscript for a second time and I am fully satisfied with the modifications and additional information brought by the authors.
I recommend publication in Nature Communications.

Reviewer #3 (Remarks to the Author):

The manuscript by Beuvier has been substantially revised and improved.

In relation to my comments, I accept that obtaining direct detailed comparison of nuclear and coccolith size is difficult. The authors correctly argue that providing this information with EM techniques is not possible since the newly formed coccoliths need to be correlated with the size of the nucleus at the time of production. However, it should be possible to provide information that supports the conclusions about nuclear size by simply measuring the size of the nucleus during the cell cycle using a fluorescent vital dye - of which there are a number of options available. Confocal sectioning should be able to provide data on the changes in nuclear size even with the smaller *E. huxleyi* cells and certainly with the larger species. Parallel monitoring of the size of newly produced coccoliths by using negative staining with calcein (staining the whole coccosphere, washing and observing the unstained new coccoliths) should be able to provide a time course of coccolith size during the cell cycle.

To take into account these remarks, the sentences of the lines 251 to 256 were replaced by :
“Even though the positive correlation between the cell size and the coccolith size was already reported within and between species⁴³⁻⁴⁴, further analysis using coherent X-ray diffraction imaging at cryogenic temperature on frozen-hydrated cell⁴⁵⁻⁴⁶ or confocal microscopy on stained cells would be needed to check whether the cell nucleus size, which varies significantly through the cell growth/division cycle, could regulate the size of *obps* and therefore the coccolith mass.”